# Diagnosing Complex Organisations with Diverse Cultures—Part 2: Application to ASEAN

Tuomo Rautakivi [1] and Maurice Yolles [2,*]

1    Faculty of Social and Political Science, Soedirman University, Purwokerto 53122, Indonesia; rautakivi_tuomo@hotmail.com
2    The Business School, Liverpool John Moores University, Liverpool L3 5UG, UK
*    Correspondence: prof.m.yolles@gmail.com

**Abstract:** In this paper, the second part of a two-part series, we explore the cultural stability of the Association of Southeast Asian Nations (ASEAN). The analytical framework adopted, formulated on a background of social cybernetics, uses Mindset Agency Theory (MAT) within a metacybernetic framework. Our exploration involves a thorough investigation of signs pointing to cultural instability, identification of potential pathologies, and the provision of insights into the underlying dynamics within ASEAN. Expanding on the theoretical foundation established in the first part, we explore the notion that regional organisations (ROs) like ASEAN can be viewed as complex adaptive systems with agency. Heterogeneity of RO membership can be both beneficial and detrimental, especially when this delivers cultural diversity. If detrimental, pathologies can arise that affect both ROs' institutional dynamics and their affiliated regional organisations, a significant interest of this paper. In response to certain cybernetic aspects introduced in part 1 of the research, MAT is shown to be a specialised framework imbued with systemic and reflexive elements. Through this, the analysis sheds light on how an agency's mindset connects with its behaviour and performance. ROs exhibit coherence in their operations when they successfully achieve adaptive goals. ROs, as agencies defined through a population of state agents, have mutual relationships and are encouraged to pursue shared regional objectives, such as economic growth, social welfare, security, and democracy. However, in highly diverse cultural environments, this poses unique challenges to achieving and maintaining cultural stability. The analysis scrutinises ASEAN's behaviour, relating it to manifestations of cultural instability, and deduces conditions that encompass an inability to undertake collective action, covert narcissism, and a lack of authority. Employing MAT as a diagnostic tool to comprehend ASEAN's intricate nature, the paper concludes with practical recommendations aimed at enhancing ASEAN's cultural health and sustainability. The ultimate vision is to foster a more integrated and proactive regional entity.

**Keywords:** ASEAN; cultural stability; Mindset Agency Theory; metacybernetic framework; regional organisations; complex adaptive systems; economic growth; social welfare; security; democracy; collective action; covert narcissism; authority; diagnostic tool; sustainability; regional entity

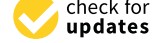



## 1. Introduction and Research Settings

This paper, the second part of a comprehensive two-part study, builds upon the theoretical groundwork of the first part [1], which establishes a theoretical framework that addresses complex adaptive systems set within a metacybernetic framework, a schematic structure constructed in a metacybernetic, philosophically rich environment that elucidates behaviour through substructural imperatives—an ontological hierarchy of intangibles that drive behavioural tendencies. In this framework, such systems can be represented as generic living systems. As explained in some detail by Yolles and Fink [2], metacybernetics is a schematic cognition–affect geometry that is context-forming using metaphor as a

means of development. There is always a possibility of connecting it with relatable commensurable approaches, or potentially those approaches that maintain at least implicitly commensurable conceptual extensions, and it operates as a theory of contexts by creating a recursive modelling process able to represent fractal situations.

Metacybernetics is closely related to but distinct from the paradigm of Complex Adaptive Systems (CAS) [3]; their connection lies in their acceptance that complex entities have agency with properties such as emergent behaviour and adaptation that are manifested from internal processes connected with interactions among a population of agents. CAS is often expressed formally through mathematical tools and applied to computational simulations, while metacybernetics has a schematic representation [4] that outlines the interconnection between an interactive tangible superstructure and its intangible substructure. The superstructure is behavioural, while the substructure, which constitutes the hidden order within the system, can be portrayed as a collective agency psyche with a multiontology. In this, deeper aspects of the complex system are revealed as the order increases. Generic living systems are minimally defined through a tri-ontology metacybernetics, often referred to as third-order cybernetics, and used to examine agency viability underpinned by an appreciation of the philosophical aspects of the modelling process.

Metacybernetics also considers the interplay between stability and uncertainty reduction, and to assist this it has adopted Frieden's probabilistic theory of information called Extreme Physical Information (EPI) [5]. One of the manifestations of metacybernetics is Mindset Agency Theory (MAT), which defines agency characteristics in terms of its substructure. These characteristics arise through formative traits, internal dynamics, and operative functionality, allowing for diagnostic insights into agency stability, coherence, and paradoxes. The MAT framework taken from part 1 of this paper is shown as Figure 1.

This figure is a cognition–affect model which recognises how the autonomous functionality of cognition and affect subagencies mutually interact, the interactions occurring through their operative systems. The model of a cognition–affect agency creates both rational cognition and emotional subcontexts. The operative system has an anterior aspect in relation to the dispositional figurative system, the two combining as an autopoietic couple, preceding the sustentative system that seeks to ensure homeostatic processes. Autopoiesis and autogenesis are higher-order process intelligences autonomously operating within both cognition and affect. The intelligences have reverse trajectories, one from a posterior system and one from an anterior system. The three types of intelligence are behavioural, operative, and figurative, all having a circular reflexive causality.

An example of a complex adaptive system is the Association of Southeast Asian Nations (ASEAN) [6]. With a specific emphasis on social organisation, this study explores ASEAN's ability to create patterns of coordination and integration among its nation-state membership. Here, a detailed investigation into ASEAN's cultural stability is undertaken. It endeavours to assess whether ASEAN exhibits signs of cultural instability and aims to identify and diagnose potential pathologies within this RO. Employing the metacybernetic modelling framework of Mindset Agency Theory (MAT), a complex adaptive system trait approach, this study explores the underlying attributes of ASEAN to determine its cultural stability. In doing so, it seeks to provide insights into the possible nature and manifestations of pathologies within the ASEAN framework.

The research question at the heart of this study centres on solving pathological problems arising from cultural diversity in complex and dynamic situations. Additionally, the study delves into understanding institutional mechanisms—expressed in terms of agency and its agents—whose characteristics are defined by a set of formative traits. By addressing these attributes, the research aims to offer a comprehensive understanding of the complex problems stemming from cultural diversity and institutional dynamics.

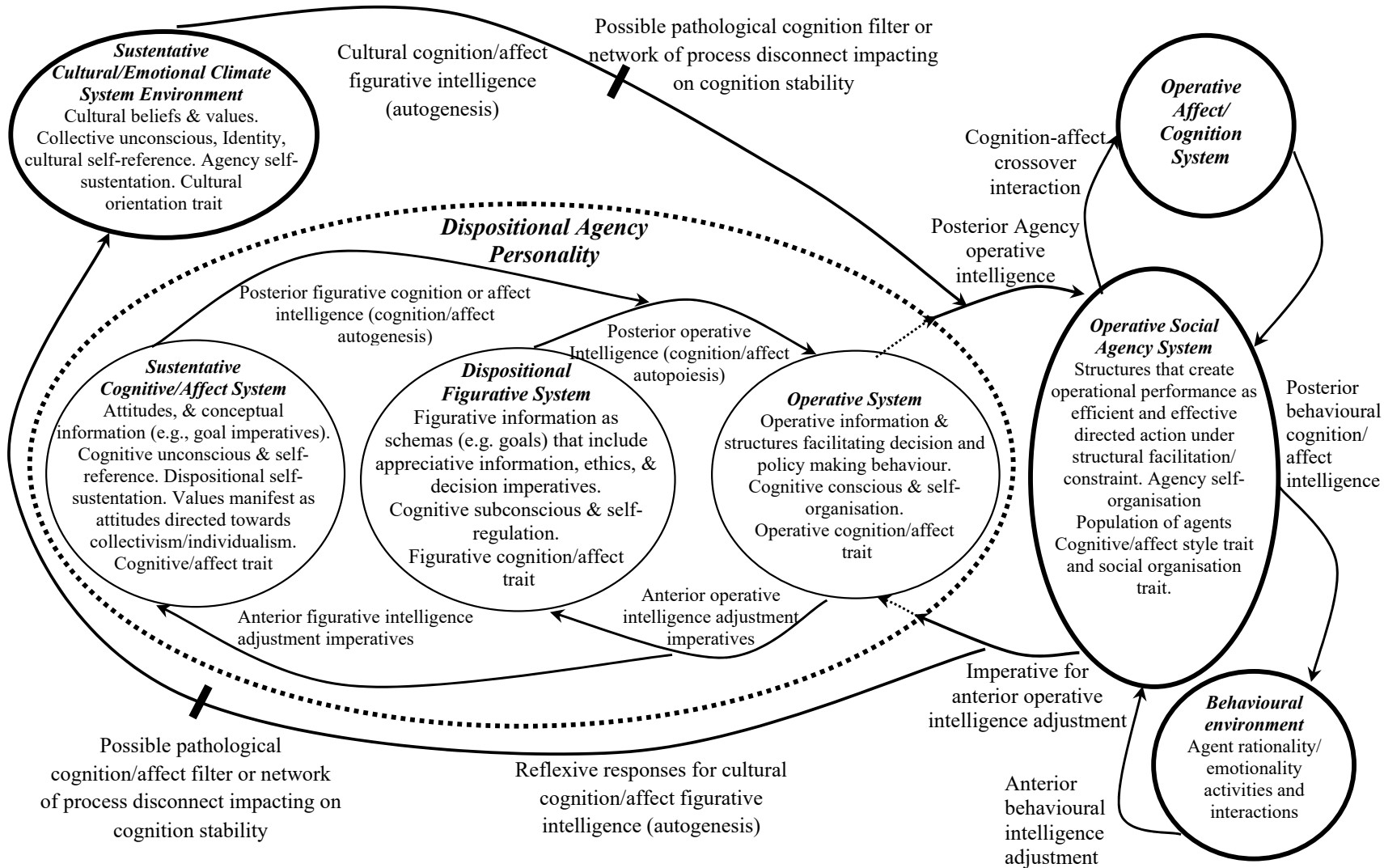

**Figure 1.** Cognition–affect model of agency (adapted from Yolles and Fink [2]).

The anticipated outcome of this study is a comprehensive understanding of ASEAN's cultural stability and potential pathologies affecting its capacity for efficacious behaviour and development. By applying MAT, the research identifies specific areas of concern within ASEAN's organisational culture, providing actionable insights for mitigating instability and addressing actual or potential pathologies. Notably, the study sheds light on ASEAN's challenges in undertaking collective action. In sum, this research contributes valuable knowledge to the broader discourse on cultural dynamics within ROs, offering practical recommendations for enhancing their cultural health and sustainability.

It was argued in part 1 of the paper that studying complex organisations is challenging because of the lack of systematic and comparative research, the fragmentation of the field, and the tension between specific and generalised methods. It was suggested that systems and cybernetic modelling can help to overcome these difficulties by organising and analysing ROs and their contexts in a systematic and reflexive way. MAT is such a model, incorporating into a prior outline framework of Cultural Agency Theory (CAT) five formative traits that create imperatives for agency behaviour. These traits can each take bipolar and contrary values that are epistemically independent. Three of these define RO disposition and two define RO sociocultural orientations. MAT elucidates how agency behaviour and performance are influenced by agency mindset. The formative traits play a central role in shaping an agency's mindset, defining its character and disposition, significantly influencing how ROs, as complex agencies, perceive, interpret, evaluate, and respond to situations, particularly in dynamic scenarios.

In order to better navigate this paper, its structure needs to be noted, recognising that its goal is to expose the underlying dynamics that shape ASEAN's cultural stability and operative coherence. In Section 2 of the paper, ASEAN's characteristics and behaviour are analysed. This analysis comprehensively examines cultural stability and operative coherence within ASEAN, scrutinising its main characteristics and behavioural patterns. To achieve this, the analytical framework, which is rooted in social cybernetics and embraced by the metacybernetic framework, acts as a conceptual metaframework. The section recognises ROs like ASEAN as complex adaptive systems and uses ASEAN's characteristics as a basis for assessing cultural stability and identifying its potential pathologies. Section 3 will explore the operative coherence and cultural stability of ASEAN, using MAT as a conceptual framework. Specifically, operative coherence is examined, recognising that incoherence may signal cultural instability within diverse contexts. In doing this, the central research question is kept in mind, which revolves around identifying and diagnosing potential pathological issues arising from the interplay of cultural diversity and institutional dynamics within ASEAN and its affiliated regional organisations. It recognises that MAT, functioning as a reflexive framework, adeptly models behaviours, performances, and the underlying imperatives that shape them. Structured into several subsections to improve comprehension, an emphasis on operational coherence is the pivotal factor preventing cultural instability across various ASEAN organisations. Finally, in Section 4, an examination of the implications of our analysis is provided. This conclusion considers the broader impact and provides insights for enhancing cultural stability within ASEAN.

## 2. The Nature of ASEAN

In this section, our focus is directed to a detailed qualitative examination of ASEAN, an RO grappling with historical challenges that, under examination, cast doubt on its competence and potentiality to harbour pathologies impeding its capacity to service its mission. To validate these concerns, we will explore the literature, and in doing this use the lens of MAT to discern ASEAN's character. The exploration will aim to provide an overall comprehension of the nature of ASEAN, and within the analysis, critical aspects such as cultural diversity, developmental disparities, organisational structure, and regional integration will be explored. The examination will extend to a historical and contextual review of ASEAN, looking at its organisational attributes and scrutinising unresolved issues related to cultural diversity, challenges, and opportunities and their impact on regional

integration and cooperation. Applying MAT to this RO embraces both cognitive and affective attributes, but also seeks to uncover potential pathologies and their consequences for performance. This holistic approach provides a thorough exploration of ASEAN's intricacies, offering insights into its functionality and potential challenges. The seamless integration of this exploration with mindset theory is intended to enrich any understanding of ASEAN's character and organisational dynamics, contributing to a detailed assessment and diagnosis of its capacity to function effectively as an RO.

The structure of this section is as follows. Initially, the background of ASEAN will provide an overview and analysis of ASEAN, focusing on its formation, objectives, challenges, and observable operational weaknesses, including its developmental disparities as an RO and the difficulties in achieving regional harmony and economic integration, as well as issues with human rights. In the next subsection, MAT will be considered as a veil through which to examine ASEAN's history and development. It will explore how cognition and affect patterns, particularly the collision between communal values and Confucian Individualism, influence ASEAN's response to challenges like the COVID-19 pandemic. The discussion anticipates potential pathologies and aims to provide insight into ASEAN's mindset and its impact on organisational functioning and stability. Section 2.3 will then examine and evaluate the cultural and mindset factors within ASEAN that may impact its ability to achieve effective and pragmatic outcomes. It will argue for a given mindset value for ASEAN (a cognition mindset of Incoherent Hierarchical Collectivism and an affect mindset of Defensive Choleric), discussing how these cultural factors may limit ASEAN's capacity for problem solving and result in vague policy proposals with neglect of practical implementation. Section 2.4 centres on member states, arguing that ASEAN tends to disregard external influences and lean towards an idea-centred approach that externally projects harmony and consensus while internally overlooking the need for problem-solving and implementation capacity. In the next subsection, cultural heterogeneity is considered, and its idealism is explored in terms of its cultural orientation. The next subsection considers the weakness of ASEAN political culture and its lack of identity, especially at the societal and elite levels. The implications of this are discussed, with illustration. n the next subsection, dispositional personality traits of ASEAN are considered, along with the resultant paradox that emerges from its behaviour, leading to failure in cooperation among its membership and lack of trust in the region. It is also explained that it is susceptible to narcissism, which leads to challenges in creating a common identity.

In Section 2.8, the much-stated meaning of "the ASEAN way" is considered, which the RO holds up as central to its being. Its paradoxical behaviour is also examined as a manifestation of this. Following on from this, the sociocognitive organisation of ASEAN is explored, highlighting its challenges due to structural weaknesses, state-centric political cultures, and a lack of enforcement mechanisms. Next, in Section 2.10, the focus is trained on ASEAN's capacity for intelligent processes that have an impact on such attributes as decision making and how its inherent pathologies can hinder policy implementation. In Section 2.11, cultural and operative anchors for ASEAN are discussed, as well as how these relates to its capacity for viable behaviour. Finally, some ASEAN institutions are examined, showing their ineffectiveness, and the reasons for this are further considered.

### 2.1. The History and Background of ASEAN

ASEAN began its regional existence in August 1967 with a meeting in Bangkok by the Foreign Ministers of Indonesia, Malaysia, the Philippines, Singapore, and Thailand and the signing of the ASEAN Declaration [7]. This defined its aims and purposes, concerned with cooperation that included economic, social, cultural, technical, and educational fields, as well as the promotion of regional peace and stability through a common respect for justice and the rule of law, as well as an adherence to the principles of the United Nations Charter.

ASEAN comprises ten member states characterised by divergent levels of development and distinct cultural backgrounds. This diversity spans from the least developed countries (LDCs) to the most developed, resulting in considerable developmental dis-

parities within the region. These disparities extend beyond mere economic inequality, encompassing variations in political structures, social development, and overall welfare. This developmental inequality is not confined to the regional level alone; it permeates into the individual states within ASEAN. The consequential differentials in development levels contribute to conditions of instability [8], where the relationship between inequality and sociopolitical instability is well established [9]. The disparities in developmental trajectories across member states underscore the complex interplay between economic, political, and social factors, all of which contribute to the overall stability of ASEAN. Addressing these disparities is necessary to engender a stable and harmonious regional environment.

ASEAN was seen to represent the collective will of the nations of Southeast Asia, and states would bind themselves together in friendship and cooperation. This would be carried out through joint efforts and sacrifices, and it was intended to provide peace, freedom, and prosperity. How central these ideas are to ASEAN must be judged by its behaviour in the face of adversity. With the ASEAN Declaration, it set up permanent missions in Jakarta, Indonesia, each mission headed by an ambassador to ASEAN who serves on the Council of Permanent Representatives (CPR), headed by a Secretary General. The council has the responsibility of local decision-making duties and coordination with the ambassadors' respective governments. ASEAN has many different working groups to coordinate efforts across different sectors and programs. Its Secretariat, also located in Jakarta, provides logistical and support services to the ASEAN working groups, representative bodies, and other ASEAN entities.

The RO has sought to improve the development of its region concerning trade and diplomacy, but it is depicted as a weak organisation in that it lacks the authority, resources, or capabilities to achieve its goals or fulfil its mission. This is reflected in its tendency to make grandiose statements that have little substance, its having no mechanism to enforce its agreements and treaties, the unintegrated state of regional banking systems and capital markets, and its member states setting their own intellectual property, land-use, and immigration policies, where there is tension over issues of cooperation and competition [10,11]. ASEAN promotes its successful intentions of improving the quality of life in the region with people-centred opportunities that collectively deliver and fully realise a capacity for human development, and this includes areas such as [12]:

(a)    Economic development plans;
(b)    Conflicts over border demarcations;
(c)    Problems with minorities within countries and border areas;
(d)    Human rights development;
(e)    Democratic development.

ASEAN is also operationally weak [10], as it fails to perform its core functions and processes effectively. Nor is there much evidence of any significant progress on these matters, and even when some changes have occurred, they have resulted in only modest outcomes. For instance, the different economies in the region remain competitive and externally oriented (with respect to ASEAN), rather than complementary and cooperative [13]; conflicts over border demarcations have resulted in little resolution, for instance, the border issues relating to Indonesia and Malaysia [14] and Thailand–Cambodia; problems with border minorities have not been resolved [15]; human rights developments have been stymied [16]; and democratic development has been stalled [17,18]. If ASEAN is to explain itself as a political body, it needs to address why it has been incapable of resolving such issues or unable to manage or develop its operations. Despite high-flying rhetoric [19], the outcomes of ASEAN's political aspirations, while claimed to be at a high level, remain at a quite low level. ASEAN member states have been traditionally described as collectivist countries [20,21]. This has been the result of surveys using Hofstede's [22] cultural values model, which has received important criticism [23], and we shall explore this further in due course. In principle, collectivist countries should be able to work well together, and we shall explain why ASEAN does not conform to this image, apparently with little ability to create collective actions.

ASEAN, as an intergovernmental organisation, is part of the public sector with its institutions, and hence it is a public organisation with a public administration. In systems like ASEAN, public administration literacy evokes negative images, and this leads to particularistic forms of decision making, a managerial euphemism for favouritism and nepotism in public organisations, and this can easily lead to a lack of confidence in and mistrust of organisations ([24], p. 58).

Perhaps because of the issues that ASEAN has, its ability to act as a coherent international strategic alliance has declined [25,26]. For Kurlantzick [27], in the 1990s and early 2000s, the ASEAN region was perceived to be one of the world's bright spots for democracy. However, after the 2010 stalled Bali III Concord, democratic and human rights issues deteriorated. On page 4 of the Bali agreement, it is stated that an intention was to: "Promote and protect human rights and fundamental freedoms, as well as promote social justice" ([28], p. 4). However, after the signing of the ASEAN Human Rights Declaration of 2012, the human rights issues deteriorated further, as illustrated by the Rohingya crises in Myanmar in 2017, the military coup in Thailand in 2014, and labour issues in Cambodia. Jones ([29], p. 79) has underscored the incapacity of ASEAN to develop by saying it "seems to be taking steps backwards rather than forward". Related issues occur in democratic development, this being on the same page as the BALI Concord III, where a statement promotes and ensures a democratic environment. Some agreements also promote economic development internal trade and intra-investment in the region: despite the agreements, ASEAN has a low level of efficacy in implementing its goals. It also has low levels of efficacy in its ability to implement actions that correspond to its aspirations and goals. The fact is that little economic importance is attached to ASEAN goals, with internal trade at around 25% and no significant changes in the last 25 years, though there has been a slight decrease in more recent years [30]. It is not only political and sociocultural factors that result in ASEAN's inefficacy in manifesting its mission behaviourally as actions. The lack of an independent character is one of the principal reasons why ASEAN is slow not only to reach agreements but also to implement them [31]. Before the passage of the ASEAN Charter, scholars had criticised ASEAN's organisational ineffectiveness due to its requirement for consensus and harmony in decision making [32].

ASEAN was constructed as a diplomatic community and was never intended to be a body for functional integration [33] and even less for structural integration with institutionalisation. That ASEAN integration is based on regionalisation means that it embraces an Asian mercantilist philosophy that favours national sovereignty and impacts the creation of institutions and institutional development. Although ASEAN has a secretariat, it is neither a decision-making body nor has it the power to implement policy decisions that are presented to it, and nor does the ASEAN Secretary General have any political power; rather, the Secretariat operates as a purely administrative bureaucracy, serving meetings.

In 1976, ASEAN adopted principles for regional stability and action, which included the creation of the Treaty of Amity and Cooperation (TAC) as a regional conflict-resolution mechanism ([34], p. 313). However, the TAC has never been implemented [35]. It aimed to promote peace and mutual respect among ASEAN members and to prevent the escalation of disputes. Later, in July 1994, ASEAN established an institution referred to as the ASEAN Regional Forum (ARF). This had two main objectives: to foster constructive dialogue and consultation on political and security issues of common interest and concern and to make significant contributions in efforts towards confidence building and preventive diplomacy in the Asia–Pacific region. It was hoped that the ARF would create a protocol and support the Dispute Settlement Mechanism. The expectation was that this would reduce uncertainty and risk by enhancing trust and cooperation among ASEAN agents, thereby freeing up resources to be used domestically. Part of its brief was to contribute towards transparency and improved monitoring of agent behaviour, while simultaneously offering increased opportunities for communication and side deals. Created to support security and sponsor annual high-level discussions within ASEAN and between ASEAN and external powers, it was set up as an informal regional body. And it was intended to

reflect principles of consensus, noninterference, incremental progress, and moving at a pace comfortable to all (called the ASEAN way). However, it lacked any binding mechanisms and enforcement capabilities to foster compliance and implementation of ASEAN decisions and agreements, relying rather on voluntary actions and goodwill. This hardly offered great incentives for conformity to decisions and agreements by wayward agents. As an institution, it was weak, having just five role positions under the special unit of the ASEAN Secretariat, with its main responsibilities being the storage, registration, and administration of ARF agreements. There was also one part-time officer within this unit, whose role was to observe and determine whether member states followed agreements [36]. Even though an edentulous organisation, it was at least an improvement for ASEAN agents who were otherwise "unaffiliated, individual countries living cheek by jowl, surrounded by major powers with competing interests in their region" ([37], p. 814).

Of the institutional bodies of the ASEAN agency, the ARF is the best known and most significant. It services a membership that includes not only ASEAN agent members, but also 10 dialogue partners (Australia, Canada, China, the European Union, India, Japan, New Zealand, the Republic of Korea, Russia, and the United States), as well as other participants, including Bangladesh, the Democratic People's Republic of Korea, Mongolia, Pakistan, Sri Lanka, and Timor-Leste, in addition to one ASEAN observer (Papua New Guinea). It functions as an instrument of security dialogue for ASEAN in the Indo-Pacific and facilitates discussions by members on current security issues. It also seeks to develop cooperative measures to enhance peace and security in the region. As such, it can act as a stabilising body in the Indo-Pacific region. The ASEAN institutional structure is agreement-centred, with agreements taking the form of declarations, as a form of ritualism. For Murray [34], these treaty-like documents are rather nonobligatory orders or EU-style directives that negatively influence the nature and efficacy of ASEAN intra-regional trade or common security policy or peace. It may be noted that the trade being referred to has not increased over the last 25 years [38]. Koga [39] explains that ASEAN is simply a set of forums where its institutional norms and rules operate, these being supported by mantras like the ASEAN way or ASEAN centrality. Thus, ASEAN draws diplomatic attention from great powers, and since ASEAN is a 10-member-state regional organisation that can (at times) speak with one voice, great powers find it attractive because, if they support what they are doing, their actions are underscored by Southeast Asian labels of "legitimacy".

## 2.2. Modelling the ASEAN Mindset

Mindset agency theory can provide a framework for understanding agency pragmatics, influenced by its perception, interpretation, communication, and adaptation to the world. Pragmatics refer to an agency's ability to cope with complexity, uncertainty, and change in its environment and to undertake practical tasks that address its needs and priorities. The theory can also explain how an agency's mindsets, which are the cognitive and affective patterns that shape its behaviour, may vary depending on the context and the parameters that define it.

Djalante et al. [40] provide a detailed investigation of ASEAN's positioning during the COVID-19 pandemic. They identify failings there which include: a lack of regional cohesiveness in regional health frameworks to develop a coherent response to the pandemic, administrative fragmentation and decentralisation, policy implementation ill-definition, an inability to adequately formulate a nonconflictual strategy, an unstable global policy initiative, uncertain relationships with health experts, shifting policy agendas, coproduction being subject to collective action challenges, legitimising policy initiatives through emotions rather than cognitions, and the description of success or failure in policy initiatives being narrative- rather than fact-dependent. As the SEAS Yusof Ishak Institute's survey 2023 report has highlighted, 49% of the respondents to the survey they undertook thought that ASEAN would be unable to recover from the pandemic [41].

This condition is a consequence of ASEAN's character as indicated by its mindset. Mindsets are determined by the values that the formative traits take, as explained in part 1 of the paper [1], which are reproduced in Table 1.

**Table 1.** Summary of traits and their agency bipolar enantiomers.

| Agency Trait | Bipolar Type | Value System Elements |
|---|---|---|
| Sustentative cultural (cognition) dimension of agency | Sensate | Sensory and material reality, pragmatism, becoming, happiness, external orientation, instrumentality, and empiricism. Tangible and concrete things are valued over abstract and intangible ones. Seeks to acquire and possess material resources and may display greed or ambition. |
| | Ideational | Super-sensory reality, morality, tradition, creation, self-examination, internal orientation, and spirituality. Cognitive autonomy. Seeks/values knowledge and understanding over tradition and authority. Learning and exploring new ideas. Curiosity or creativity. |
| Sustentative cognitive dimension of dispositional personality | Intellectual Autonomy | Individual uniqueness, expression, meaning, and independence. Individualistic. Self-reliance/autonomy. Values own opinions and interests. Can challenge or ignore the norms and expectations of others. |
| | Embeddedness | Social relationships, identification, participation, shared goals, order, tradition, security, and wisdom. Collectivistic, social harmony/equality. Group membership and identity are valued, involves cooperation or compromise with others for the common good. |
| Figurative dimension of dispositional personality | Mastery + Affective Autonomy | Self-assertion, mastery, direction, change, monism, egocentric or altruistic ends, and meaningfulness. Self-assertion. Opinions/ feelings confident and open. Seeks to influence/persuade others. May display dominance or leadership. It links to Affective Autonomy which concerns emotional well-being, excitement, and variety arising from within—an inner drive or motive for positive feelings. |
| | Harmony | Understanding, appreciation, pluralism, unity with nature, environmental protection, and world peace. A tendency to accept and adapt to situations without resistance or complaint. Seeks to maintain peace and balance, may display tolerance or flexibility. |
| Operative dimension of dispositional personality | Hierarchy | Hierarchical roles, obligations, rules, authority, legitimacy, power, and benefit of the organisation. Conformity, accepting and following norms and expectations of an agent's social position/status. Agents seek to fulfil and perform roles, and display loyalty/obedience. |
| | Egalitarianism | Moral equality, co-operation, concern, choice, negotiation, service, and welfare of everyone. A belief that all agents have equal rights and opportunities regardless of social status or role. Agents seek to promote fairness and justice and may display solidarity or empathy. |
| Operative social dimension of agency | Dramatising | Interpersonal events, communication, narrative, belief systems, social contracts, individual benefit, and ideocentric agencies. Interagency relations. Tendency to focus on and enhance self-interest and benefit through action-oriented and expressive behaviour. Agents seek to attract and impress others and may display charisma or dramatisation. |
| | Patterning | Configurations, curiosity, collectivism relationships with individuals, allocentric. Concerned with social relationship configurations. Has a tendency to form and maintain complex and diverse social networks based on collective benefit and action delay through observation. Agents seek to optimise and coordinate their social interactions and may display pragmatism or strategizing. |
| | Gesellschaft | Modern, urban, and impersonal societies focusing on Individualism and pursuing agency interests. Agents seek to adapt and innovate in their changing environment and may display independence or ambition. |
| | Gemeinschaft | Traditional, rural, and collectivistic communities with a strong sense of loyalty and shared values. Agents seek to preserve and honour their cultural heritage and may display devotion or reverence. |

ASEAN is superficially a Patterner RO, since relationships and coherence are said to be extremely important to it, as is the goal formation that it deems to be for the benefit of its collective membership. However, beyond these words, the actual relationship between its agents is Dramatist, as we shall explain shortly. It is culturally Ideational in that it supports pragmatism with an externally related orientation, and its interest in greater integration does not extend to the creation of mechanisms that can facilitate this. Its disposition, recognised as a collective personality, may be understood by initially referring to the ASEAN slogan indicated earlier of "One Vision, One Identity, One Community", and this highlights a Gemeinschaft sociocognitive organisation that is underpinned by Collectivist values and is theoretically comfortable with a Patterning trait value. The problem is that its interagent relations are problematic because the agents function in a way that satisfies self-interest and individual benefit. Such fragmentation does not sit well with the idea of ASEAN having "One (personal) Identity", so that its public identity becomes a false self, indicating an identity schism. Pragmatically then, ASEAN operates with an incoherent sociocognitive style, resulting in agency instability. This suggests that it is not capable of delivering pragmatic outputs that relate to the events that impact it. It also appears that the ASEAN personality is essentially Individualistic, though its sociocognitive organisation is one of Gemeinschaft. As already suggested, this could create issues that result in uncommitted Collectivism due to the inherent contradiction between personality imperatives and their operative social orientation. However, it must be said that its Individualist personality is Asian, which puts a particular stamp on its character. To explain this, Safitr [42] recognises that ASEAN embraces the Asian values of Confucian ethics, in which harmony, unity, and community come first. She also includes consensus in this, but consensus bears a similarity to Confucian harmony [43], which is conditioned by the important Confucian dedication to hierarchy. Thus, to deal with hierarchy, Asian cultures have developed their own manifestation of Individualism. This is illustrated by Brindley [44], who explains how Confucian Individualism does not stress an individual's separation, total independence, and uniqueness from external authorities of power, as tends to be adopted in Western Individualism. Rather, it centres on an emphasis on power relationships as connected to unity (or harmony) with external authorities of power. Confucian Individualism, unlike Western radical individualism, provides an agent with a holistic integration with the authoritative forces that exist in its agency environment. The agent is recognised as a significant integrated component of agency, where individual values, empowerment, authority, control, creativity, and self-determination have individualised attributes.

These attributes are represented by the cognition mindset of Hierarchical Collectivism (determined by Intellectual Autonomy, Mastery + Affective Autonomy and Hierarchy) from Table 6 in part 1 of this paper [1]. This is a Collectivist mindset, quite different from the Individualist mindset of Hierarchical Individualism (determined by Embeddedness, Harmony, and Hierarchy). The differences between these two lie in the agency traits allocated (which are either Patterning, Dramatising, Sensate, or Ideational). For instance, the distinctions between the Individualist mindsets are highlighted by the differences in individualism in the West and Asia. Western individualism might be typified as, say, the cognition mindset of either Hierarchical or Egalitarian Individualism, depending on the dominant agency political ideology. Confucian individualism is both relative and relational, giving agents the freedom to make their own decisions in a global agency, thereby shaping their own trajectories within the complexity of the existential interactive interrelationships. This gives agents the authority to satisfy their potential while negotiating environmental influences, commands, and responsibilities. This results in an agency–authority tension that many see as a paradox. It may be seen that this tension occurs in ASEAN. So, despite the promotion of its motto that supports a Gemeinschaft sociocognitive organisation, its Confucian Individualism collides with its collectivist values, creating figurative intelligence pathologies, so that it fails to implement them either strategically or operatively. A likely association with the cognition mindset of Incoherent Hierarchical Collectivism is the affect mindset of Defensive Choleric, with its dispositional affect traits of containment, protection,

and dominance and with missionary and empathetic cognition traits. ASEAN's interest in protection is illustrated by its report into fiscal matters characterised by a variety of measures that include: liberalisation intended to improve national investment, facilitation to ease administrative needs concerning fiscal and business matters, promotion through support by information flows and facilitation agencies, and regulations to enable an improved fiscal environment [45]. It also seeks to become a dominant regional influencer [46], consistent with the ideas of Zheng Guoxiang [47], who notes that Confucian independence is also subject to the extensive responsibilities and obligations that exist in a network of relationships. This illustrates the inseparable relationship between the individual and the community, whilst highlighting the independent personality and achieving a distinctive self while penetrating the community. The self creates a relationship that is both internal and transcendent to society.

ASEAN is Patterner-oriented, where key attributes are configuration and personal relationships, where allocentric collectives are important, and where members operate subjectively and are culturally Ideational. Hence, ASEAN is Ideational, unconditionally embracing moral positions and creating an environment with the potential for increased integration. The figurative system shown in Figure 1 enables perception to result in mental imagery. It can provide preferred ideological images that may facilitate action; this is located in the operative system (hence, Egalitarianism), which provides the ability of an agency to implement values in action [48].

While post hoc analysis like this is very illuminating and useful to understand the capabilities of an organisation, understanding its collective mindset can suggest likely issues with its sociocognitive organisation from which issues can be anticipated, enabling the potential for resolution. To illustrate how mindsets can be used in this way, we shall accept that ASEAN has a cognition mindset of Incoherent Hierarchical Collectivism and an affective mindset of Defensive Choleric and show that its behaviour is consistent with these interactive mindsets. Summarising ASEAN traits, recognising that we have defined two operative traits, one cultural trait, and three personality/dispositional traits, we obtain the following:

ASEAN Agency Traits for the Cognition Mindset of Incoherent Hierarchical Collectivism:

1. Agency of cultural Ideationality: Idea-centred rather than pragmatic, unconditional morality, supporting tradition, a tendency toward idea creation, and self-examination.
2. Dispositional personality of cognitive Intellectual Autonomy: Supports notions of autonomy/uniqueness among agents, expresses internal attributes (like feelings), and independently pursues ideas/intellectual directions.
3. Dispositional personality of figurative Harmony: As a pluralistic organisation, agents pursue their own ideas and intellectual directions independently, with mutual understanding and appreciation (not exploitation), unity with nature, and the world at peace.
4. Dispositional personality of operative Hierarchy: Power is hierarchical, normally unequally distributed, and supports a chain of authority.
5. Agency sociocognitive style is incoherent. This means that while its social relationship structure is Gemeinschaft, its actual cognitive style is Dramatism. This suggests instability in its autopoietic processes, making it problematic to create adaptive requisite responses to complex changes in its environment.

ASEAN Agency Traits for the Affect Mindset of Defensive Choleric:

1. Agency cultural emotional climate that is Missionary: The imposition of ideas on others and the conversion of others to and the heralding and promoting of ideas; susceptible to propagandism and revivalism.
2. Dispositional personality affects containment: dependability, restraint, self-possession, self-containment, self-control, self-discipline, self-governance, self-mastery, self-command, moderateness, and continence.
3. Dispositional personality of figurative protection: safety, stability/security, protective shield, safety, conservation, insurance, preservation, and safeguarding.

4.  Dispositional personality of operative dominance: control, domination, rules instituted through supremacy/hegemony, power, pre-eminence, sovereignty, ascendancy, authority, command, and susceptibility to narcissism and vanity.
5.  Agency of socially operative empathy: accepting, compassionate, sensitive, and sympathetic.

Consideration had been given as to whether, rather than fear-oriented, ASEAN might be security-oriented, which is a function of trust. However, according to Roberts [49], the frequency of interaction throughout the region has not strongly influenced the level of trust in each of the ASEAN agents.

While these characteristics anticipate behaviour, they do not predict pathologies. These depend on the self-producing stability of ASEAN and its capacity for self-stabilisation. This in turn depends on its network of processes that define its operative and figurative intelligence, as shown in Figure 1, where pathological filtering of figurative intelligence can be responsible for an inability to self-stabilise and a pathological filter on operative intelligence is responsible for strategy–operations stability.

The paradox that typifies ASEAN makes this RO a prime candidate for deeper exploration. Thus, in the next section, we shall examine ASEAN to determine whether its behaviour is determined by the proposed traits that depict its character.

*2.3. Mindsets and ASEAN Performance*

ASEAN recognises its agents as a group of interdependent and unequal members who value loyalty and stability over change and innovation and who are assertive, confident, and goal-oriented but tend to lack sensitivity and flexibility in dealing with others. Consider that ASEAN has a cognitive mindset of Incoherent Hierarchical Collectivism and an affective mindset of Defensive Choleric. These cultural factors may limit its ability to achieve effective and pragmatic outcomes, as it focuses more on following procedures than solving problems [29]. Therefore, it often generates vague and unrealistic policy proposals, while neglecting the practical aspects of implementation and evaluation [50].

Consider that ASEAN has a cognitive mindset of Incoherent Hierarchical Collectivism and an affective mindset of Defensive Choleric. Due to cultural issues, this may be indicative of instrumentality, so that it is only capable of making "process not progress" through nonpragmatic trajectories [32]. At present, vague policy ideas are relatively prolific, but pragmatic policy initiatives (i.e., policy details and processes of implementation) reside at some distant, inaccessible horizon [50].

One way to assess the performance of ASEAN as an RO, as it seeks to promote developmental improvement through economic growth, social progress, and cultural development among its member states, is to use the concept of pragmatics. While pragmatism is concerned with the ability of an agency to undertake practical tasks, pragmatics enhances the concept by referring to the agency's ability to cope in its behaviour with complexity, uncertainty, and change in its environment [51]. Successful pragmatics can be measured by applying the criteria of development evaluation [52,53], which considers the relationship between agency intervention and the context of that intervention, and can be used to determine the meaning and value of such intervention [51]. The criteria are relevance, efficiency, effectiveness, efficacy, sustainability, and impact. Relevance means how well the intervention addresses the needs and priorities of the ASEAN member states and the challenges and opportunities in the region and beyond. Efficacy refers to the ability of ASEAN to produce a desired result. Effectiveness means how well the intervention achieves the objectives and outcomes of the ASEAN agreements and decisions and whether they conform to the ASEAN Vision 2025 [48,54] and the ASEAN Community Blueprints [55]. The ASEAN Vision 2025 outlines the aspirations and goals of ASEAN for the following decade, while the ASEAN Community Blueprints outline the goals, strategies, and actions for each of the three pillars of the ASEAN Community: political security, economics, and socioculture [56]. Efficiency means how well the intervention uses the available resources and capacities of the ASEAN institutions and mechanisms. Sustainability means how well

the intervention contributes to the long-term development and integration of ASEAN and to its peace and stability. Impact means how well the intervention creates positive changes and benefits for the ASEAN member states and for the region.

In addition to these criteria, we also introduce efficacy, which refers to the pragmatic attainability and feasibility of achieving an intervention, given the constraints and opportunities of a given context. Efficacy relates to the potential and capacity of the ASEAN institutions and mechanisms to implement and deliver interventions, as well as to the alignment and coherence of any interventions concerning ASEAN values and principles. Efficacy also reflects the responsiveness and adaptability of the ASEAN institutions and mechanisms to changing circumstances and emerging issues. Efficacy can be seen as a precondition for effectiveness, efficiency, sustainability, and impact, assuming relevance, as well as a criterion for evaluating the quality and value of the intervention (cf. [51]). This is so since efficacy refers to the extent to which the intervention is attainable and feasible given the constraints and opportunities of a context. If an intervention is not feasible or attainable, then it cannot be effective, efficient, sustainable, or impactful, regardless of how relevant it is. Efficacy may also be seen as a criterion for evaluating the quality and value of the intervention since it relates to the potential and capacity of the ASEAN agents and mechanisms to implement and deliver interventions, as well as to the alignment and coherence of any interventions with ASEAN values and principles. Efficacy also reflects the responsiveness and adaptability of ASEAN agents and mechanisms to changing circumstances and emerging issues. Such aspects of efficacy enable the assessment of the successfulness of an intervention in relation to any standards and expectations of ASEAN.

Alternative regimes might be used to evaluate ASEAN with respect to its pragmatic outputs. These should reflect agency learning and adaptive capacity, at least through inquiry and reflexive considerations [51], where learning enables prediction, problem resolution, and pragmatic action (cf. [57]). Pragmatics can be seen as an important aspect of assessing ASEAN's performance as an RO because it captures how well ASEAN responds to its complex dynamic environment and its capability to achieve its intended outcomes. Here, we shall not concern ourselves with the development evaluation criteria as such but will be interested in preconditional efficacy that would permit further analysis to occur. To examine the efficacy of ASEAN performance, we shall reflect on our mindset model through qualitative arguments from the literature. As we shall see from this, ASEAN's capacity towards pragmatics will demonstrate significant inefficacy. In a survey by Choi et al. [58], 83% of respondents thought that ASEAN is slow and ineffective and thus cannot cope with fluid political and economic developments, thus becoming irrelevant in the new world order.

The potential for ASEAN to fail as an RO can be seen in terms of its cultural heterogeneity. This is the result of a diverse culture within and among ASEAN member states and can be a major source of instability in the region. Following Huntington [59] and Rosa, Penna, and Carvalho [60], cultural heterogeneity may become a catalyst for ethnic and religious tensions, humanitarian crises, and political and economic challenges, leading to clashes and, in some instances, precipitating humanitarian crises. This has implications for the cultural heterogeneity of ASEAN's vision of building a peaceful, prosperous, and integrated community. Such ASEAN instability reflects the diversity and complexity of ASEAN's member states, with their different histories, cultures, systems, goals, and challenges.

The evident prominence of strong national cultures and identities overshadows and weakens the development of a cohesive ASEAN common culture and shared identity, with member states exhibiting a willingness to emphasise and promote their distinctive national cultures. This results in an ASEAN culture with weak norms, and such cultures likely exhibit cultural instability and lack of integration ([2], pp. 47–48). How this relates to ASEAN will be further explained. Other aspects of ASEAN's cultural system, like its structure and its process intelligences, will also be explained.

### 2.4. ASEAN and Its Agents

ASEAN essentially disregards external influences, despite its member states signing up to international agreements and laws. This is a classic example of closed-system behaviour, as explained by Nulad [61], where she notes that ASEAN member states stipulate that domestic laws can trump universal human rights. This constitutes an extreme level of state-centrism.

ASEAN has a population of agents, each with its own ideology. Collectively, this conforms to some form of Asian mercantilist economic policy with the idea of harmony and consensus following ASEAN statements and concords. As such, ASEAN is an idea-centred organisation, rather than an organisation with problem-solving and implementation capacity. As such, there is no mechanism to inhibit the creation of conflicts and obstacles that may arise where decision making is to be manifested in operations. Lin [31] observes that while many ASEAN agreements are technically binding, the implementation of these agreements relies on the voluntary compliance of member states. This is because ASEAN agreements lack enforcement mechanisms for ensuring that member states adhere to the measures outlined in the agreements, thus creating a situation where there are no repercussions for noncompliance. ASEAN has no central institutions, power, or authority to uphold compliance and force action. Nor do the ASEAN agreements force member states to do anything; rather, they recommend what ASEAN states "shall" do, and the statements it does make are more like intentions than agreements. Thirdly, as Kurlantzick [17] observes, in the ASEAN Charter 2007, ASEAN did indeed draft and sign a new charter in 2007, but it maintained most of the ideals of consensus and nonintervention from the original ASEAN Declaration. Rüland ([62], p. 439) explains that "ASEAN's collective identity is crystallized in the revered principle of non-intervention". Though the new charter did commit to creating a "just, democratic, and harmonious environment in the region", it did not define any of these terms and contained no provisions, as exist in other ROs, for agents to intervene in the affairs of other agents, for instance, in the case of gross abuses of human rights [17]. Later, the 2011 Bali Concord III referred to the promotion of human rights, democracy, economic cooperation, and disaster management, but still there is no definition of what human rights and democracy mean or how to measure and define disasters. So, agents only have recourse to interpreting and implementing the statements independently, possibly leading to contradiction and conflict. Lin ([31], p. 836) notes that ASEAN leaders lack explicit legally binding provisions in most of their agreements and have no effective compliance mechanisms or credible dispute settlement systems. Further, ASEAN does not often carry out measures already agreed on to integrate the regional economy or deal with transnational problems.

As already noted, human rights and democratic development have even declined. Following a working paper of the Council on Foreign Relations [63], ASEAN was not able to create more coherent and interdependent economic ties between its agents, for example, to assist with the Philippine typhoon catastrophe in 2013, nor was it able to offer practical help to find the missing Malaysian Airlines flight MH370 or lead the missing aeroplane rescue operations (noting that information came from Australia, not from ASEAN). ASEAN's basic orientation is consensus with harmony, and Harmony arises from figurative orientation. Harmony is pluralistic, and within it, one tries to understand and appreciate rather than to direct or exploit. Harmony-oriented organisations base their ideas on the notion that the world should be accepted as it is and understood and appreciated (where the possibility that it needs to change is not a consideration), whether organisations aim to direct or exploit the environment or render it static [48]. Contrary to a Mastery orientation, which promotes the idea of assertiveness and control over the natural and social environment to achieve personal or group goals, the Mastery approach emphasises values such as ambition, success, daring, and competence. Organisations with a Mastery orientation are typically dynamic, competitive, and goal-oriented, often utilising technology to manipulate and shape their environment to achieve success and desired outcomes. These orientations arise from figurative traits, including cognitive and cultural traits. Harmony is associated with

Collectivism. ASEAN countries are represented by Incoherent Hierarchical Collectivism from mindset theory ([48], p. 41), being harmony- and idea-centred, and tend to embrace the *creation* of ideas [64]. However, individuals who generate ideas often struggle to put those ideas into practice due to a lack of practical skills or the necessary resources to bring their ideas to life within a system. Organisations with a primarily Ideational mindset focus on exploring and developing a wide range of ideas, placing less emphasis on how to implement them tangibly. As a result, they may create diversity but struggle to effectively utilise and apply it ([48], p. 31).

On the other hand, an excessive emphasis on harmony may lead to a lack of motivation to take action. In such cases, minimal progress is made in response to survival challenges, and the enjoyment of nature may be limited by an inability to address natural threats. While harmony can promote social cohesion and enhance collective achievements, an overemphasis on harmony may hinder overall progress and adaptation.

### 2.5. The Cultural Diversity of ASEAN

The member states within ASEAN exhibit a spectrum of diverse cultures, showcasing cultural heterogeneity. This diversity extends beyond merely distinct cultural practices, encompassing variations in religions and even different chronologies (historical timelines, periods, or sequences of events that have shaped the cultural development of each member state), each contributing to the unique cultural fabric of individual member states. The cultural orientation within these states is inherently influenced by political and social agencies, shaped within the ambient host culture that encapsulates them. Within this web of cultural diversity, cultural anchors emerge, representing the paradigm embraced by each agency. These anchors serve as foundational elements within the agency's cultural framework, fostering the development of both formal and informal norms [65]. These norms dictate patterns of behaviour, modes of conduct and expression, as well as forms of thought, attitudes, and values. The adherence to these norms varies among the agency's membership, creating a dynamic interplay of cultural influences within ASEAN's diverse and evolving cultural landscape.

We have deemed that ASEAN has an Ideational cultural trait. While agencies may take cultural traits that are Ideational or Sensate, following Sorokin [66], the traits are locked in an interactive dynamic embrace that can generate an outcome that enables one or other of the two traits to dominate but where the other trait may have a sufficient presence to make an impact. ASEAN, however, is dominated by its Ideational trait, this being illustrated by its ability to generate ideas that it is unable to implement. This lack of pragmatics unconditionally supports the creation of ideas, morality, and tradition [67]. Its Ideational force operates beyond its normative underpinnings and plays a significant part in its self-maintenance. This trait affects its notions of regional integration and provides explanations concerning its collective identity, which can always potentially provide an influential approach to the analyses of subjective issues [68]. As an illustration of its Ideationality, Cambodia (if taken as a representative agent of ASEAN) supports balance, stability, and harmony, and this is achieved through moral and social control, tradition, and conformism [69]. Moral positioning is also an attribute of ASEAN within its "ASEAN way" with respect, for instance, to its position on human rights and its duties towards community, where public morality plays a part [70]. The idea-centred Ideational culture is often unable to apply and then implement its ideas in action, and it may lack the practical capabilities or material governing controls necessary to manifest the ideas behaviourally [71]. Ideational culture is also considered to be important for achieving harmony in society and maintaining a static and stable equilibrium ([50], p. 57), and this is closely related to a harmony-oriented society.

ASEAN has a loose culture with: a low degree of normative conformity and a lack of coordination among its agents, with a low level of accountability and legitimacy regarding its mechanisms and institutions; many sources of diversity and variability in its norms, such as different agent political systems, cultures, religions, languages, and interests; many sources of disruption and deviation from its norms, like disputes over borders, resources,

and sovereignty; humanitarian crises; economic disruptions; and security threats. It projects a culture as an integrated identity framed through discourse that is delivered beyond the region of Southeast Asia, but this creates only an illusion of substance ([32], p. 149). Its culture is also passive, since the values it espouses are not pragmatically manifested in action. The nature of the cultural trait is that it determines what type of leader it appoints, what laws are created, and what rules are imposed and policed. The values that determine the trait are reflected in the political culture, which consists not only of the norms and values but also the beliefs and knowledge that include the rules and procedures and rituals that it relies on [72]. These components are formulated as operative intentions, where all agents interpret rules and values as procedures from their perspective, and this can change over time and with situational change. This does not define a strong or common ASEAN political culture that drives common ASEAN political behaviour and procedures. This is not surprising recalling Jones and Smith's realisation that ASEAN political culture is substantively illusory and has only a set of competing agent cultures and no dominant influence to determine how they may work together as a whole.

ASEAN agency is also deemed to have a cultural emotional climate with a missionary trait, involving the imposition of ideas on others and the conversion of others to and the heralding and promoting of ideas and susceptibility to propagandism and revivalism. The imposition of ideas on others also appears to be a characteristic of ASEAN, as illustrated in Vietnam, where managers tend to apply executive power according to the missionary trait, thus influencing technical, communication, and information flow processes [73].

ASEAN member states are traditional top-down societies, and under normal circumstances, through the legitimisation of selected patterns of behaviour, top-down influences can constrain the nature of interactions at the lower level [74]. However, such constraints by legitimisation may become ineffective in situations in which there is uncertainty, especially where crises arise [75,76].

Organisational culture determines how laws (which are longer-term social regulators) and rules (the result of shorter-term political regulators) are implemented and acted upon [77]. The legal formality of ASEAN does not specify any legal rights to do anything. Rather, it requires its 10 dialogue partners to sign agreements individually when they are made on behalf of ASEAN, in a way similar to FTA agreements. This is in contrast to the EU, where it can sign as a unitary agency on behalf of its agent members to ratify agreements.

The fact that ASEAN does not function adequately as an independent unity is one of the principal reasons why it is slow not only in reaching agreements but also in implementing them [31]. The ASEAN mercantilist and state-centric ideology, through figurative intelligence, can represent the cultural belief system (of values, attitudes, and beliefs) as a coalescence of normative ideological and ethical standards of the culture that ultimately defines what it is that constitutes legitimate modes and means of behaviour [78]. This leads to the situation in which ASEAN agents are not willing to adopt legal power for its control processes, thus diminishing their capacity to manage and direct their sovereign status. ASEAN leaders also lack explicit, legally binding provisions. This has led to a situation where ASEAN statements are more like political communiqués without legal status and where commitment need not be followed with implementation and action. The statements are intentions of what should be done, written in conditional forms, rather than commonly accepted agreements of what must be done collectively.

*2.6. The Failings of ASEAN Political Culture*

Political culture is constituted through the political values and norms that a political organisation adheres to. The political culture may be strong when the values and norms are strongly manifested in strategic and behavioural attributes of the organisation or weak when they are not. The political culture of an RO will affect its degree and scope concerning the kind of political integration that is possible, and political institutions will also affect (i.e., reinforce or change) the values of the political culture. Political culture and political institutions affect each other and have interrelated connections [8,79]. As noted

earlier, ASEAN member states have been concerned primarily with state building rather than the building of an RO. States in a region that together build within a mercantilist philosophy may also limit the level and efficacy of regional processes and the creation of regional institutions.

Naturally, this impacts the efficacy of ASEAN performance, and, as a result, it suffers from weak state regionalism [80]. ASEAN identity remains quite weak within ASEAN states at all layers of society, even among the elites [81]. Identity is key to building a community, whether economically, socioculturally, or in terms of political security. It is important to highlight that the identity statistics of ASEAN exhibit significant variations depending on the source, and controversies arise, particularly concerning questions related to the identity of Southeast Asians. Notably, Southeast Asia is a geographical region, while ASEAN represents an institutional framework. For Ayoob [82], a distinct subaltern (a social/political marginalisation within a social hierarchy) realism observed among weaker states ultimately seeks to foster national identities rather than regional ones, emphasising a focus on individual nationhood within the broader social and political hierarchy. ASEAN's public identity, which relates to a larger social environment and reflects social roles, norms, expectations, and obligations, is also weak among its member citizens. It is influenced by cultural factors and other social categories like national cultural traditions and history [83]. This is one factor that contributes to ASEAN's cultural heterogeneity, with problems arising in creating a common culture and identity where "the cultural clue" is missing. The covert reason for the creation of ASEAN was an external reason and the threat of communism. Anti-communism gave a sense of common identity to states, so that they were collectively referred to as the "Balkans" of Asia ([19], p. 348). ASEAN has difficulty in achieving cultural homogeneity, but cultural characteristics may create cultural conflicts and a state-centric approach or even state fetishism.

Bosteels [84] refers to a term that describes the strengthening of the state's central power, where it prioritises the interests of private and transnational companies. As weak regionalism or weak member states, embracing harmony to support the notion of noninterference in other member states can be seen to be devoid of the potential to create an effective platform for social coherence between member states and their people. Despite ASEAN regarding itself as the most successful organisation in Asia since its inception 50 years ago [85], its achievements in the region during its existence leave a lot to be desired. Since ASEAN has a general lack of interest in closer or "substantive" direct political and economic integration for its agents, cooperation and a shift towards integration have occurred without any institutional frameworks [86]. ASEAN leaders and national ruling elites have not shown any interest in creating institutional frameworks that enable the creation of an Asian superpower or a major national power [87]. ASEAN integration is at best shallow, proportional, or conditional. Also, ASEAN declarations, charters, and agreements are written without specific meanings and definitions of issues. The ASEAN model is typical of other regional cooperative organisations operating through norms and statements. The mechanism is divergent, as with other regional cooperative organisations. A distinctive difference is the mechanisms (procedures and principles) of cooperation and the decision-making process. As already noted, ASEAN is a diplomatic community with private and informal procedures that seek to avoid institutionalisation.

There is thus considerable evidence to support the realisation that the ASEAN regional community is weak and that it can account for very few of ASEAN's actions [17,18,88,89]. Since its inception, ASEAN has not shown itself to be relevant and may even be classed as a permanently nascent community with lots of unrealised potential. Its own principles and political culture and the ASEAN way are obstacles to its taking coherent action, as shown in the latest Myanmar military coup d'état that occurred in February 2021. After the coup, ASEAN stated that there is "dialogue, reconciliation and the return to normalcy" in Myanmar while it cited the principles of democracy of the ASEAN Charter [90]. The coup also demonstrates the value of the principles of the rule of law, good governance, human rights, democracy, and constitutional government in the ASEAN Charter [90]. There is

little reason to think that most ASEAN states will respect these commitments [27,88], since ASEAN charters are not obligatory but are rather statements of aspirations [88] with no mechanisms provided for manifesting these. Seng [90] notes that member-state agents are left to manifest the values and principles of the Charter concerning their establishment, implementation, and preservation while ASEAN follows its principle of noninterference. When Thailand was informed by Myanmar that its coup d'état was a domestic internal issue and that it needed to resolve the problem on its own, the ASEAN organisation was quiet. Similarly, Cambodia's Prime Minister, Hun Sen, gave statements indicating that Cambodia would not comment on a country's internal issues, following the ASEAN basic principle of noninterference. Optimistically, Malaysia just hoped for peaceful negotiations. ASEAN had a similar response to Myanmar's Rohingya crisis, while the international community expected more than this from ASEAN [90]. If the RO could confront new challenges, then this could lead to a new framework of activities. Regarding this, the Indonesian Foreign Minister, Retno Marsudi, noted that to do nothing is not an option [90]. ASEAN's future accreditation in the international arena will depend on how it can handle such current issues. Pongsudhirak [91] predicted that Myanmar's putsch will likely become a lose–lose outcome for ASEAN credibility and centrality. He noted that, similarly, ASEAN was quiet about Thailand's earlier coup d'état. He noted that, so far, ASEAN's efforts have been unimpressive.

Since the onset of COVID-19, ASEAN member states suffered from the global pandemic at the same time that Myanmar had its political problems. The RO's collective action in response to COVID-19 was controversial, though Tan [92] notes that while ASEAN's response to the pandemic was underappreciated, relatively few data were obtained from member countries. Tan also notes that ASEAN agents are stepping up to cooperate substantively during a crisis, but there is still a problem of lack of information access and communication. The problem here is that there is an inadequate sharing mechanism in ASEAN, which leads to a lack of robust information that can mitigate the RO´s collective effects. Despite this, Kliem [93] sees the situation differently, noting that the ASEAN region has done reasonably well in its response to the pandemic. As a caveat to this, he explains that ASEAN has been unable to match the resolve of its member states, and that there is a substantial gap between timely and robust national pandemic management and inadequacy at the regional level.

According to Almuttaqi [94], the ASEAN regional grouping appeared sluggish in developing a regional response to COVID-19 and instead adopted what he described as a nation-first mentality. He criticises the member states for acting independently for their own interests rather than for ASEAN's collective interest. Nandyatama [95] recognises an underlying problem due to the lack of shared information among the nation-state agents and the problem of ASEAN leadership inadequacy, since it does not have a leading country to provide leadership. In addition to the leadership requirement, Mattli [96] notes that there are three preconditions needed to enable successful cooperation and integration. The first of these is undisputed leadership, such that the region must have a leading country which serves as the focal point in the coordination of rules, regulations, and policies. The other two are a strong market pressure for integration, where there is significant potential for economic benefits, and provision by an integration treaty for the establishment of committed institutions, such as centralised monitoring or third-party enforcement, which helps to catalyse the integration process. Mattli has noted that it is impossible to underestimate the institutional role in fostering regional integration, the European Union RO having all three preconditions. However, ASEAN has only market pressure, and even this has not significantly developed over the last 25 years.

Earlier, we noted that ASEAN has a weak degree of cohesiveness, and as Buendia [97] notes, interstate relations and regional cooperation consist of avoidance of formal mechanisms and legalistic procedures for decision making and a reliance on consultation consensus to achieve collective goals. In an extension of this, Nandyatama [95] underlines that ASEAN never responds collectively to any regional crises when they occur, but, cre-

atively, is more willing to formulate a new ASEAN mechanism after a crisis has passed. He gave a similar example of the Chiang Mai Initiative (CMI), noting that the bloc's legacy from the 1997 Asian financial crisis was only formed after 2000.

Probably the best example of ASEAN's weak political culture and level of efficacy is shown through the South China Sea dispute and the creation of a Code of Conduct (CoC). The South China Sea dispute has a history that begins with the ASEAN Ministerial Meeting in Jakarta in 1996, where the Manila Declaration was reaffirmed [80]. In 2002, Peking's comfort with the ASEAN process culminated in 2002 in the signing of a nonbinding Declaration on the Conduct of Parties in the South China Sea, and that declaration reaffirmed China's five principles of peaceful coexistence [80]. A weak constitution of boundaries and a loose membership in the ASEAN framework may have a beautifully designed façade but very weak foundations (cf. [98]) that lead to ASEAN diplomatic limitations in the South China Sea dispute [80]. Gamas [99] explains the weakness of ASEAN consensus and principles as ASEAN ritualism. Here, the role of ASEAN political culture is shown in the case of the CoC, which resulted in a lack of consensus in the 2012 biannual ASEAN summit, chaired by Cambodia, which concretised ritualism rather than providing a clear pragmatic statement. He explained ASEAN's failure in the 2012 summits in Cambodia to provide a cohesive platform among its members and produce a binding CoC. This was due to the underlying political culture in Southeast Asia [99], despite the talk of unity among the members. Because of the absence of unity and coherence in ASEAN or even solidarity between member states, ASEAN came to suffer the effects of weak state regionalism. Both the Philippines and, more remarkably, Vietnam looked increasingly to the United States when confronted by China's renewed assertiveness [80]. The Southeast Asia 2023 Survey report data showed similar results and even the increasing incoherence of ASEAN [40]. The result shows that 61% of respondents saw that ASEAN is becoming increasingly disunited. The interesting point of this survey was that the respondents represented ASEAN elites, like academia, the business and finance sector, government, and regional and international organisations.

Later, at the ASEAN Summit in Singapore in April 2018, the RO was again unable to manage the South China Sea dispute when a divided ASEAN rather than a strong collective RO was shown to be in effect. Kurlantzick [27] observed that the ASEAN 32nd summit took the same pattern as previous ASEAN summits, with a traditional consensus style that hampers the possibility of addressing issues. He noted that public statements made during the summit were meaningless, since any language that could be construed as critical had been eliminated. After the summit, ASEAN was still unable to develop a position on the CoC in the South China Sea and instead began negotiations with China on a code [27]. Heydarian [100] notes that it has been more than twenty years since the idea of a code of conduct had been raised and 15 years since the signing of the (nonbinding) Declaration on the Conduct of Parties (DoC). However, ASEAN was still in the middle of what some see as a never-ending negotiation. The never-ending CoC story is also reflective of the ASEAN Agreement on Disaster Management and Emergency Response (AADMER). Since its appearance in 2005, AADMER has not been able to assist in the resolution of problems. It is better seen as more of a surveillance and observatory group organisation, rather than an organisation with implementation skills and action capacity, and its financial base comes from voluntary fees. ASEAN catastrophe aid is based on bilateral aid rather than ASEAN RO aid, with an illustration provided by South Thailand's floods of 2017. Malaysia wished to assist Thailand, but ASEAN did not. The ASEAN structure with its harmony orientations does not favour action, and it prefers to make statements and provide ritual outcomes.

### 2.7. ASEAN Dispositional Personality

The disposition of ASEAN hinges on figurative mental models and abstractions that have been solidified from the strategic parts of the agency of ASEAN. Here, the cognitive trait of Intellectual Autonomy strongly supports autonomy/uniqueness among agents, expresses internal attributes (like feelings), and independently pursues ideas/intellectual directions.

ASEAN offers a paradox that results from contradictions in its processes, permitted by its polity [101–104]. Thus, in the context of regionalism and integration, the ASEAN paradox arises through the tension between the logic of regionalism (shared norms, values, and interests underpinning regional cooperation) and the limits of integration (challenges that arise from the diversity of agent interests and priorities) [105]. The goal of ASEAN was to preserve long-term peace based on intergovernmental talks, without formal regional institutions, preferring a purely decentralised system. ASEAN members have agreed on a set of procedural norms which have become the principles of "the ASEAN way" [106]. These constitute a set of working guidelines for the management of conflicts that occur within the boundary of ASEAN. Norms lead to cooperation among states but not to the establishment of institutions following the basic idea of mercantilism. However, ASEAN is not very effective in creating cooperation among its member agents. This is because while it is good at generating ideas that conform to its ideology, its inherent contradictions deliver paradox. These contradictions arise due to the informality of ASEAN [107], which has grown more fractured through its inability to deal with conflictual situations like the civil war in Myanmar and the admission of Papua New Guinea as a member, and where trust across the region is extremely low [108]. As an example, regarding security issues, ASEAN generates contradictory/paradoxical rather than pragmatic solutions [102] concerning terrorism in the region, and there are no mechanisms in place to deal with this [101].

Its figurative system is deemed to have a Harmony trait, and as a pluralistic organisation, its agents pursue their own ideas and intellectual directions independently, though there is a supposed mutual understanding and appreciation (rather than exploitation) and a search for unity and peace. Its plurality is reflected in the varying components of its different ethnic groups [109], but that plurality is heterogeneous, with variations in the institutions and regional political security based on the divergence of agent political cultures and historical experience ([110], p. 2). As an RO, it adopts a principle of mutual understanding, predictability, trust, confidence, and goodwill among member agents [111]. The idea of agent appreciation within RO plurality arose historically with Asia's Buddhism which, while a minority religion in ASEAN, is a major factor there [112] that promotes principles that seek to enhance growth potential, provided the content of growth reflects the broad principles of sustainability and nonexploitation [113].

The operative personality trait of ASEAN is Hierarchy, where power is hierarchical and normally unequally distributed and a chain of authority is supported. Hierarchical values also support the legitimate unequal distribution of prosperity [55]. ASEAN operates through a hierarchical power structure [114]. Power is also centralised and concentrated, and it is unequally distributed; for instance, a global leader and their subordinates working in Malaysia might rarely "think outside the box", and the subordinates would expect to be told what to do. They are also, therefore, individually less innovative and avoid speaking to their bosses directly, especially with controversial positioning [115].

This brings us to the affected personality, the trait of which is deemed to be containment. It involves a need for dependability, restraint, self-possession, self-containment, self-control, self-discipline, self-governance, self-mastery, self-command, and both moderateness and continence. As Antolik [116] explains, ASEAN was a product of the combination of common fears and weaknesses rather than common strengths, and so to foster group solidarity, its leadership has adopted three tactics. The first of these is to stress the virtue of dependability, followed by an incremental approach to decision making and the promotion of community consciousness. Also, as a representation of ASEAN positioning, "moderateness" is a hallmark of Thailand [117].

Its affect figurative trait is deemed to be one of Protection, oriented towards safety, stability/security, the creation of a safeguarding protective shield, safety, conservation, insurance, and preservation. It has already been said that the mission of ASEAN is to maintain political security in its community and to provide for its well-integrated economics and a socioculture that enhances the quality of life among the citizens of its member states [118]. These are underpinned by its values, which may be identified as "respect, peace and secu-

rity, prosperity, non-interference, consultation/dialogue, adherence to international law and rules of trade, democracy, freedom, promotion and protection of human rights, unity in diversity, inclusivity, and ASEAN centrality in conducting external relations" ([119], p. 1).

Finally, the ASEAN operative trait of personality is deemed to be Dominance, involving control, domination, and the production of rules that are given to supremacy/hegemony, power, pre-eminence, sovereignty, ascendancy, authority, and command. There is also a susceptibility to narcissism and vanity. Hegemony, as a part of dominance, refers to the ascendency or domination in an RO agency of one of its agents over another and can be argued to be an alternative approach to hierarchy in regional governance, but, according to Misalucha [120], ASEAN hierarchy is projected as a benign hegemon, where dominant authority over others is applied in a benevolent or harmless way so that there exist multiple types of regional rule that provide a demonstration of ongoing efforts by agents towards building and maintaining deeper relations with each other.

While ASEAN may be susceptible to narcissism, its overt/benign form is self-serving and manipulative, while its pathological form is also malicious and creates maladaptive efforts for self-regulation. Pathological narcissism is likely to be seen when an identity schism occurs, and it is conceptualised by the two features of narcissistic grandiosity and narcissistic vulnerability, where the former refers to specific deficits in interpersonal functioning and the latter to vulnerability as associated with all forms of dysfunction [121]. Benign narcissism may be seen to occur in ASEAN as a "narcissism of minor differences", which describes its tendency to exaggerate the difference between it and others [104]. There is a connection between the narcissism of minor differences and the narcissistic personality. From a political perspective, certain political orientations, for instance, are represented by forms of populism, differentiating between "us and them", where the "them" are held to be in some way inferior in our context, and this exaggeration is essentially a narcissistic position. The Freudian notion of narcissism of minor differences explains rivalry amongst people with common ties and, more broadly speaking, amongst neighbouring states, where there tends to be a focus on minor differences in order for states to define their "uniqueness" and thus their identity. ASEAN recognised the pivotal role of ASEAN identity in community building at the ASEAN 37th summit 2020, but still they have a problem creating a common identity. This relates to "the ASEAN way" (that recommends sensitivity, avoidance of narcissism, and knowing one's place ([122], p. 389)) and which is a decision-making approach blind to alternative positioning concerning the cultural perception of the radical nature of the word "no", leading to its official exclusion. This exclusion limits the possibility of regional growth in terms of member states or diversity, and Timor-Leste does not fit into the "Asian profile" because of its European influences, its democratic system, and its human rights records [104]. If it is perceived that ASEAN is susceptible to corporate narcissism, then an analysis must move beyond tangible attributes to its intangible corporate personality profile, seeking to identify any pathologies that might arise therein. One of the indicative signs of narcissism is self-contradiction [123], which ASEAN is guilty of [124]. Other attributes are personality characteristics like excessive or grandiose self-importance, entitlement, exploitation of others, and a lack of ability to understand or care about others—the latter perhaps being reflective of ASEAN's position concerning minorities like the Rohingya.

The cultural system includes self-identification information and functions as a self-stabilising/homeostatic control that regulates the relationship between the substructural metasystem and the structural system. This involves values and norms which facilitate the development of strategic structures like goals, ideologies, and ethics in the figurative system. Self-regulation defines and formulates goals, standards, and motivations toward identifiable outcomes [125,126] like the ASEAN Economic Community (AEC) of 2015. Without defining information or self-regulation, no progress can occur and development is difficult or even impossible. Thus, Kurlantzick, ([17], p. 4) notes that "although ASEAN vowed to form one 'Economic Community' by 2015, including a single market and production base, it likely will not realize that goal". Nor is there any detailed information on or

definition of what the ASEAN Economic Community means. Benny et al. ([127], p. 5) note that "Regarding the concept of Economic Communities, a review of the literature found no specific definition of it despite the many kinds of economic integration". The ASEAN Economic Community was a goal intended to come into being on 1 January 2015, but this date was then reset to 4 January 2016. There are still unsolved problems, like goods or products of origin, and how to determine the origin of a product. So, ASEAN was able to create the notion of the ASEAN Economic Community but was unable to respond to the issues that arose with its creation. Another problem is the origin of information, especially where "digger" information becomes available, as illustrated by laws only being available in local languages. As is revealed on the ASEAN Secretariat's website, there is no disclosure of any internal law that governs the being of member states. If there is no information and interaction between member states of ASEAN, it is difficult or even impossible to find real information about what, for example, a researcher needs, there being no common language. Also, government offices are not willing to give any information to outsiders from inside the organisation. ASEAN defines its agreements in wide frameworks without clear and exact definitions but seems to interpret and implement economic agreements with little coordination with its member agents [128].

*2.8. The ASEAN Way as an Attitude*

The ASEAN way is an attitude constituted as a principle and hence functions as a code of conduct which has become the basis for the mindset of Incoherent Hierarchical Collectivism. The attitude is a manifestation of ASEAN culture and is contrary to pragmatism, cementing ASEAN's commitment to Ideationality. ASEAN was constructed as a diplomatic community [81], and its weak political culture is underpinned by the norm of noninterference in the affairs of member states. This constitutes "the ASEAN way", which Jones and Smith [32] define as the process through which intermember interactions occur, through a process of discretionary cultivation, informality, expediency, consensus building, and nonconfrontational bargaining. The ASEAN way also includes an Asian mercantilist approach, where the sovereignty of international institutions is weakened even though formal ASEAN political institutions exist in theory [106]. The philosophical base that underpins ASEAN does not create a favourable platform for institutional development or the creation of a strong ASEAN political culture. This norm is consistent with a general tendency in Asia for interactive processes that are nonconfrontational, with the avoidance of open disagreement between discussants. This is underpinned by the ASEAN principles that offer a code of conduct to govern interstate relations in Southeast Asia, stated in the Treaty of Amity and Cooperation from 1976, which is defined as: (1) respect for member-state sovereignty and territorial integrity; (2) noninterference in internal and issues and politics; and (3) the settlement of disputes by peaceful means and the renunciation of the threat or use of force [106].

The goal of ASEAN was to preserve long-term peace based on intergovernmental talks without formal regional institutions, the preference being for a purely decentralised system. ASEAN members have agreed on a set of procedural norms that have become the principles of the ASEAN way [106]. These constitute a set of working guidelines for the management of conflicts that occur within the boundary of ASEAN. Norms lead to cooperation among states but not to the establishment of institutions following the basic idea of mercantilism. The ASEAN way states that the principle of noninterference is the original core foundation upon which regional relations between ASEAN member states are based [129]. Biziouras [107] has described the ASEAN way as an informal, consensus-oriented decision-making process. Antolik [116] notes that this level of decision-making flexibility has been deemed necessary in creating a regional structure that has not assumed initiatives that are not fully and wholly supported by its member agents, thus increasing the chances of the survival of the regional organisation. As a counter to this, ASEAN needs to carry out actions rather than aim at ends. Koga ([130], p. 91) has made a strong statement about the ASEAN way in saying that it promotes an excuse for relegating ASEAN to a "talk

shop", which Webber [131] has concurred with in noting that it offers high-flying rhetoric. Koga [130], recognising how contexts may change, notes that the ASEAN Way is a means for other ends.

An illustration of the ASEAN way can be seen in the RO's approach to issues of security, which appear to be most commonly seen in terms of contradictory, or rather paradoxical, positions. Leifer [102] is interested in ASEAN regional peace and security, recognising that it has not instituted a structure that is capable of fulfilling this need. Davies [101], in his discussion of security in relation to terrorism in the region, notes that there are no mechanisms in place to deal with it. Hazri [103] is concerned with the problem of the Rohingya, the Myanmar military government having been accused of genocide against the country's Rohingya minority [132], and while a more integrated ASEAN organisation is being sought, groups like the Rohingya are peripheral entities that are disconnected from government-to-government affairs and are not of interest to ASEAN. Another paradox, one that does not concern security but rather membership eligibility, is considered by Sefixas et al. [104] in their examination of the case of Timor-Leste. This small state expressed its desire to join ASEAN in 2008. It was admitted "in principle" as the organisation's 11th member, but its full membership is pending [133]. Its difficulty in joining appears to be because of a perception in ASEAN that it is more European than Asian, despite broadly satisfying the membership criteria. This presents an issue of paradox that centres on ASEAN's "narcissism of minor differences", which, as already noted, describes an agency's tendency to exaggerate the difference between it and others [104]. It signifies a differentiation that ignores important differences and pluralities among those in favour of differentiation based on trivialities, these being perceived as a threat to the sense of self of the narcissistic personality ([134], p. 184).

The ASEAN way is the second principle of noninterference for ASEAN [135], which exists together with a state-centric approach. This provides a weak platform upon which to build a strong and coherent connection between ASEAN agents, diminishing any ability to act as a global player in the international arena. The resulting incoherence [136] still occurs after half a century, and it is still an obstacle to closer cooperation; it is nothing new. The same territorial conflicts still occur between Thailand and Cambodia, Cambodia and Vietnam, the Philippines and Indonesia, Indonesia and Malaysia, and so on. There are also minority problems in almost all ASEAN member countries, and they have increased rather than decreased. Personal disputes between leaders have occurred between Malaysia and Indonesia. Conflicts related to the Cold War were solved, but they were not resolved by ASEAN; rather, they were resolved by the collapse of the Soviet Union and other external events. Incoherence also affects ASEAN unity and its capacity to create a common security policy, as the South China Sea dispute shows. While Vietnam and the Philippines protest against China, aggression continues in the region, like that of the Spratly Islands and the Philippines Exclusive Economic Zone (Reuters, 2018). In these cases, ASEAN was unable to create any protection or show any solidarity or unity with respect to Vietnam and the Philippines. From an analysis undertaken by Kurlantzick [27], the ASEAN summit of 2018 produced little substance on important issues like the South China Sea. Despite territorial problems, ASEAN was unable to give any statements about the South China Sea dispute or show any coherence and unity regarding its member states. Geopolitical pressure hit ASEAN member states, but ASEAN left Vietnam and the Philippines to stand by themselves. Many ASAEAN members are dependent on China and other external players.

Heydarian [89] explains that ASEAN proposed a CoC in the year 1996 for long-term stability for the region. It has taken three years for ASEAN to submit a proposal to China. China agreed to the declaration of the Conduct of Parties in the South China Sea in the year 2002. It has taken a while for ASEAN to be able to respond to its own declaration in any way. The ASEAN summit in Singapore in 2018 had similar results to previous summits concerning the South China Sea. Kurlantzick [27,48] explains that ASEAN removed any language that could be critical. He also notes that it is unclear whether ASEAN member states will be able to unite to develop a relatively tough common position on a CoC in the

South China Sea, having just begun negotiations with China on a code. All these actions from the past to the present time show a low degree of coherence, solidarity, and unity between its member states.

*2.9. The Sociocognitive Organisation of ASEAN*

In part 1 of this paper, consideration was given to the nature of sociocognitive organisation through the relationship between Gemeinschaft/Patterning and Gesellschaft/Dramatising, and here it is useful to relate this more closely to ASEAN. In its social cognition trait of Patterning, social and other forms of relational configurations occur, with social influence in dynamic relationships, persistent curiosity, symmetry, pattern, balance, and collective goal formation being important, as are subjective perspectives. ASEAN is also classified as having a Gemeinschaft sociocognitive organisation, operating through collective structural relationships with collective goals and understandings, and its agents are connected by shared customs and traditions [137].

Sociocognitive organisation is influenced by the affect cultural agency trait, which is the dominant emotion that defines the agency emotional climate. The trait therefore also influences how its agents act and interact with each other and can have either direct or indirect influences on these aspects. Direct influences are those that are clear and immediate, such as the expression, communication, or regulation of emotions. For example, the cultural agency affect trait may influence how the agents display, convey, or control their emotions. Indirect influences are those that are subtle and mediated, such as emotional norms or external emotional factors. For instance, it may influence the emotional norms and values that shape an agency's sociocognitive organisation by influencing its hierarchy or decision-making process. The trait may also influence the beliefs and attitudes that affect both an agency and its agent behaviour and interagent relations, such as trust, cooperation, and conflict. Thus, by interacting with the figurative affect trait it can influence emotion regulation that has consequences for an agency's rules, laws, and policies. The cultural agency affect trait may also internally influence institutions, as well as external actors that interact with the agency, such as its allies or rivals.

For ASEAN, this emotional climate trait is fear, and it can have both positive and negative effects on its agents. It can make them isolate themselves, avoid cooperation, and feel insecure, anxious, and aggressive [138]. Fear can also make them act defensively or pre-emptively [139], and this can trigger conflicts among agents [139]. Fear can also lead to aggression when agents face high levels of perceived threat or danger from others [138]. However, fear can also motivate agents to seek cooperation and security through collective action and mutual support [140]. Fear can also foster mutual empathy and understanding among those who share similar experiences and challenges [138]. ASEAN was founded as a trust-building mechanism for mediating disputes between its members [141] rather than as a platform for mediating disputes [142]. It has successfully reduced interstate conflict by adhering to principles of consensus, noninterference, and peaceful resolution of disputes [142]. However, these principles have also faced limitations and challenges in addressing new and complex issues and crises in the region and beyond [143]. The many meetings and informal social gatherings of ASEAN create interpersonal trust, and this enables disputes to be addressed without resorting to formal legal mechanisms. However, the approach adopted prevents the creation of effective interventional mechanisms in interagent conflicts which are deemed to be domestic issues and therefore not a concern for ASEAN. It is also unable to handle interstate disagreements which cannot be resolved on the side-lines.

The structure of ASEAN is different from that of other regional organisations and institutions. It cannot force member countries to comply with agreed regulations because there are no mechanisms for this, and there is an absence of sanction clauses and political power or authority, as well as a weak and only informal means by which disputes can be resolved [128]. Moreover, ASEAN does not have the authority to enforce human rights, cannot manage natural disasters, and has no mechanisms for conflict resolution [128]. Such

structural weaknesses generate a lack of confidence in the organisation or trust in its ability to pragmatically manifest goals.

ASEAN member states can be characterised as countries that are traditionally state-centric within their political culture [144]. By state-centric is meant that the state is of central importance and state sovereignty is undisputed. A state-centric approach together with a harmony orientation and noninterference principles are a weak basis and platform to set ASEAN as a strong and coherent unit between member countries, and it is still further away from being a global player in the international arena. This unfortunate combination of factors that contributes towards its inefficacy has been referred to positively as "the ASEAN way". This positivity simply permits political pathology that negatively affects ASEAN procedures and operative systems to be brushed aside with an empty phrase that distracts one from recognising reality and validates that negativity. So, rather than being in a stage of improving development concerning its aspirations, ASEAN may well be in a state of declining development. When ASEAN engages in intergovernmental discussions, the role of the state and its sovereignty needs to be implicitly considered, as well as its potential for interference in domestic member issues. ASEAN member states are characterised as countries that are traditionally state-centric, and this may even extend to state fetishism [135], which highlights the central power of the state. The state-centric approach is embedded in the ASEAN hybrid governance system that underpins its loose, weak, and passive culture, one that allows its values to create an agency anchor but does not actively participate in strategic or operative functions concerning knowledge processes, learning, or creativity [145,146].

*2.10. ASEAN Intelligences*

Agency operates through various process intelligences. Referring to Figure 1, we remember that cultural figurative intelligence is an agency's ability to represent cultural values and beliefs by merging normative ideological, ethical, and behavioural standards, which in turn signify social legitimacy [74]. The effectiveness of this intelligence enhances system viability, while inefficiency hinders it. Therefore, efficacious cultural figurative intelligence can help moderate conflicts and ultimately enhance system viability among multiple competing cultural groups. Figurative intelligence enables the development of appropriate policy instruments that align with an agency's ideology and ethics to address its surroundings. It encompasses a network of processes that involve a set of figurative images, including mental models and abstractions.

On the other hand, normative agency personality refers to an agency's ability to select and pursue its interpretation of a fulfilling life ([147], p. 45). Normative agents are required to adhere to the standard concepts associated with artificial agents and possess the ability to represent norms in a way that allows for reasoning and modification throughout the agent's lifespan, including knowledge representation, learning, morality, and law [148]. Normative agent architectures are primarily rooted in belief, desire, and intention [149].

The figurative system focuses on generating understanding and formulating goals derived from data collection and a careful evaluation of arguments rather than impulsive decisions driven by decision makers' spontaneous desires ([150], p. 10). While figurative intelligences play a crucial role in influencing effective decision making, they may be susceptible to pathologies that hinder an agency's ability to implement policies.

Such a pathology can be shown in the case of ASEAN. Thus, Kurlantzick [18] criticised Concord III concerning its incapacity to promote human rights, facilitate economic and democratic development, and establish processes of disaster management. It did not help that ASEAN did not offer definitions of what these things meant to it. Similarly, criticism of the ASEAN Charter 2007 can be applied to the notion of a democratic and harmonious environment in the region, which had not been defined, and which contains no provisions to enable state members to intervene, as exist for other regional bodies ([17], p. 5). An absence of such definitions also denies the creation of a measurement system, including measures of outcomes [31]. This interconnected issue between definitions and

measurement is important, for what is to be measured and how? The *Herald Tribune*, on 15 December 2008, went as far as to say, "Up until now, the 10-nation organisation has been little more than a talk shop, forging agreements through consensus and steering away from confrontation". The Council on Foreign Relations in 2014 also noted that, despite the statements about human rights and democracy, the fact is that both sectors are In decline.

It has been noted that ASEAN has high aspirations in producing statements of ideas and declarations but that it offers little evaluation of those statements [151]. Smith further notes that the ASEAN Bali Concord II of 2003 was a recursive (or, one might say, regurgitated) statement that was regenerated from old ones. The same problem occurred with the ASEAN statement in 2007, which repeated ASEAN's original declaration instead of creating a new charter with new ideas [17]. Indeed, ASEAN has not implemented measures and often does not carry out measures [31], nor does it provide detailed strategies or time frames to implement plans. A strategy should arise from figurative intelligence and be pragmatically formulated through operative intelligence. ASEAN's capacity to implement new ideas relates to an Ideational culture that emphasises maintenance and understands and supports traditions and traditional systems rather than exploiting new ideas [66].

### 2.11. ASEAN Instrumentality

We have noted that instrumentality occurs in an agency when it has no anchors that enable it to maintain homeostatic self-sustainability. The anchors have both a cultural and an operative dimension. The cultural dimension is the most important because this is where knowledge, values, and norms are maintained, and it operates as a meta-self-regulative (or self-sustaining) influence that engenders homeostatic control of the agency. When culture is weak or loose or in any other way passive, it does not provide the controls necessary for the network of figurative information processes that enable the agency to maintain homeostasis. The agency also requires a representative voice, and this is dependent on a coherent culture. This is because a coherent culture can create a sense of trust and collaboration among agents, enabling them to express their views and concerns freely and constructively, and engenders agency relevance. A representative voice may also emerge as a result of a shared vision and common values that guide the agency's actions and decisions. This does not mean that a coherent culture will always result in a representative voice, especially if there are factors that influence agency culture. Such factors may include environmental complexity and diversity, including different political systems, cultures, religions, languages, levels of development, and interests. In the case of ASEAN, another factor might be the principle of noninterference in the internal affairs of any of the agents, and this limits the scope and depth of cooperation and dialogue on sensitive issues. Yet another factor might be the fact that decision making is consensus-based, requiring unanimity among all member states; this may result in delays or compromises. The influence and interests of external powers, such as China, the United States, India, Japan, and Australia, might also be factors and may affect the unity and autonomy of ASEAN. Its member states have different relationships and ties with different external superpowers, and this can more easily lead to conflicts of interest and geopolitical pressures, since ASEAN has no common policy or strategy or foreign policy. A representative voice may arise from different mechanisms in ASEAN. For Inama and Sim [128], a strong executive is required for ASEAN to implement and enforce its agreements and decisions and to coordinate and supervise the various ASEAN bodies and mechanisms it has. Such an executive would require certain aspects of agent sovereignty to be delegated to ASEAN, the nature of which would be bounded by prior specifications. However, Inama and Sim note that the ASEAN Secretariat is currently too weak and understaffed to perform such functions effectively. Rather, it needs to establish a supranational executive body that is adequately resourced and has sufficient power and accountability, as has the EU.

Whether the executive is strong or weak, it may be subject to pathological autopoiesis [152], damaging its potential for viability and autonomy, and may lead to poor performance, loss of trust, resistance to change, and an inability to adapt to changing environments.

This is also typical of an instrumental organisation. A weak executive is more prone to pathological autopoiesis, as it may lack the authority, resources, or accountability to monitor and correct the agency's behaviour and culture. It may also be more influenced by internal pressures or interests that reinforce the agency's self-image and worldview. The weakness of ASEAN is shown by its lack of a central authority to speak on its behalf with its agents to regulate processes, achieve agreements, or conduct actions. Seemingly consistent with a condition of autopoietic pathology, Jetschke [124] explains that ASEAN continues to use rhetoric that declares its intention to enhance cooperation and devise projects the materialisation of which lies at some distant horizon. As an illustration, it has been devoid of major institutional innovations with limited levels of interagent interactions.

As an instrumental system, ASEAN has very few norms that are not shared and has not engendered a sense of unity in the face of transboundary threats, such a condition leading to a weak institutional structure [32] and a lack of structural cohesion. ASEAN faces a lack of integration and hence cohesiveness among its member states, and this impacts communication and information flows and the potential for policy making and creates fragmented responses to issues. ASEAN's weak degree of cohesiveness ([97], p. 5) highlights that it has significant issues concerning its regional unity, given the cultural variation across the region, its ethnic diversity, the distinct stages of economic development between member states, and the variety of political systems practised in Southeast Asian countries. This also implies a reduced capacity for ASEAN to develop any improvement in its institutions. An institution may be considered to assume connected values and structures, the latter formulated as collections of formal and informal norms, rules, procedures, protocols, sanctions, and habits and practices (cf. [153]), all of which contribute to behavioural coherence and the intersection of the workings of social norms. To develop improvement, institutions require evolving generic political structures and behaviours and conventions or norms.

The nonintervention norm of ASEAN together with its harmony orientation and its weak political platform severely limits any processes for regional integration. It should be noted that the Myanmar coup d'état on 1 February 2021 is a classic example of ASEAN's incapacity to handle domestic issues. Consistent with the noninterventionism of ASEAN agents, Thailand's deputy prime minister, Prawit Wongsuwanin, has noted that Thailand does not express a view on this issue and that it is an internal one for Myanmar [154]. Consistent with this position, the prime minister of Cambodia, Hun Sen, said that they do not comment on any internal issues of any other country [155]. Similarly, Malaysia only highlighted the importance of a peaceful settlement, while Singapore and Indonesia only sought to follow the situation. Such positioning follows the principle of harmony between member states and of noninterference in internal and domestic issues and politics. Before this conflict, neither ASEAN nor its member states commented on Thailand's coup d'état in 1999/2000.

Cohesive organisational groups, with common dominant values and norms, are better able to deliver information and generate normative, symbolic, and cultural structures that impact agency behaviour, according to Granovetter [156], who also states that (structural) embeddedness refers to the fact that economic action and outcomes, like all social action and outcomes, are affected by dyadic/pairwise relations and by the structure of the overall network of relations. A mindset dominated by a Hierarchical Collectivism personality would satisfy this, being led by the Embeddedness trait, with dominant Ideationality and Harmony. Embeddedness is enabled in extended kinship and social trust network contexts through a Gemeinschaft sociocognitive organisation. The social trust network (called Guanxi in Chinese) develops as nonfamilial relationships are transformed into familial ties with a growth in interpersonal trust that enables the progress of complex transactions [157]. ASEAN's sociocognitive style is incoherent because the Hierarchical Collectivist personality is operatively connected in the RO, with a mismatch between the Gemeinschaft social relationships and the Dramatist cognitive style, thereby creating instability. As such, it has become an instrumental agency. This means that its identity is undeveloped or inaccessible

to its operative capabilities and that it lacks self-organisational and adaptive capacity, resulting in reduced viability. Reduced viability means that the agency has a lower ability to maintain its existence and functionality in different situations and contexts. It also means that the agency is more vulnerable to environmental changes, internal disturbances, and external challenges that might threaten its survival or performance. Additionally, it means that the agency has a lower resilience, robustness, or sustainability in different situations and contexts. This is because it is effectively controlled by any residual strategies that it might have, the selection of which is likely determined by external forces or influences. Residual strategies are the strategies that the agency has left over from its previous identity or viability but that are no longer necessarily relevant or effective for its current situation or context. Residual strategies might also be postulated strategies that the agency assumes or pretends to have but that are not based on any evidence or reality. They may also be strategies that are empty of meaning. Here, agency has no understanding or appreciation of them, and they are imposed or expected by others, since it cannot manifest operationally any goals or values it might have. This can mean that it is unaware or indifferent to its situation and environment and that any cognitive–affective processes that occur have no impact on its responses to changes in its environment. Such agencies may also be referred to as cleaver zombies.

While ASEAN may be an instrumental agency, its member agents will likely not be instrumental, since their cultures create anchors for their developmental processes. As a reflection of this, we can consider the cultural trait values of some illustrative agents and ascertain how small differences in trait values can result in mindsets with minor mindset variations and hence behaviours, thereby explaining to some limited degree the distinctions that can be observed between the different characters of the ASEAN agents. For instance, consider the impact of small shifts in the cultural trait values between Ideational and Sensate cultural traits for Ideationally dominated societies [158], recalling that Sensate cultures are more focused on empirical evidence and material facts, while Ideational cultures are more focused on spiritual or religious beliefs. Such respective orientations will contribute to variations in agent characteristics. When expressing differences, it must be recognised that all the considered nation states (as agent members of ASEAN with their dominant Ideational trait values) have aspects of Sensate culture that provide some degree of mixing, as the Ideational and Sensate values interact through the daily sociopolitical and economic behaviours of individuals. This mix, according to Sorokin [66], can move towards Idealistic trait values (a fusion of Ideational and Sensate cultural trait values), which only occurs as the culture takes a Sensate trajectory, where Ideational and Sensate values maintain a balance.

### 2.12. ASEAN and Dispute Settlement

The ASEAN Dispute Settlement Mechanism is incapable of solving any controversial questions or issues related to economics and trade, security, or border disputes among ASEAN member agents. Perhaps this is because, as Hsu [159] explains, it exists nowhere other than on paper. To give it real functionality, Hsu suggests that its location should be moved from Jakarta to Geneva. It has already been noted that ASEAN prefers external bodies to resolve issues that it is connected with. This is supported by Sim (2008), who notes that international issues between ASEAN countries are currently resolved through the World Trade Organisation (WTO) or the International Court of Justice (ICJ). The absence of dispute resolution mechanisms in every aspect of the ASEAN way has already been noted, and this impacts the possibilities of cooperation. A good example is the Phra Wihan Temple border problems between Thailand and Cambodia. ASEAN was unable to address the issue, and the International Court of Justice made a decision (effectively on its behalf) in November 2013. In another case, Indonesia and Malaysia disputed Pulau Ligitan and Pulau Sipadan in 2002, and there was also the case of the Malaysian and Singaporean issue over polyethylene in 1995. Like the Dispute Settlement Mechanism, many of the so-called ASEAN institutions exist only on paper (Jones, 2010) or in theory [106,160]. Jones [35]

sees ASEAN's institutional capacity and its existence critically. ASEAN has no range of internal mechanisms to foster political cooperation, financial cooperation, or sociocultural cooperation. It does, however, have internal–external mechanisms to foster economic cooperation like AMRO or CMIMM, and this means that force and pressure come from outside of ASEAN [161], and it is this that drives its behavioural responses.

ASEAN has a figurative orientation that arises from its preferred position of stability and acceptance of situations as they are rather than direct exploitation of them. ASEAN does not have any effective behavioural and operative intelligence due to the underdevelopment of its structure and its weak supportive institutions, which therefore makes it unable to adequately support collective actions. Weak states create weak institutions that in turn create weak institutional and political structures. Where these states are members of an RO like ASEAN, they operate independently, seeking their own interests in a way that is likely to be devoid of collective interest. The independent states operate by the intergovernmental system, but without the ASEAN political culture. ASEAN also does not include political control, political direction, or any control system, which leads to a weak institutional structure [32]. Institutionalisation makes organisations more than just an instrument to achieve certain purposes [162]. Referring back to Figure 1, the pathology filter ASEAN has on its figurative intelligence is indicative of a culture that is either weak or passive or both, such that it simply operates as an instrumental agency. We recall that a weak culture has core values that are not clearly defined, communicated, or widely accepted by those working for the organisation, and where there is little alignment between the way things are done and the espoused values, this leads to inconsistent behaviour. We also recall that a passive culture is one in which cultural aspects do not actively participate in strategic or operative functions concerning knowledge processes, learning, or creativity, and it is, therefore, unable to provide a stabilising mechanism to enable its autonomous decision making. Thus, in such a case, individual agents respond to events that are significant for ASEAN through the local anchors of their own agent cultures rather than any common ASEAN culture.

Following Chilton [163], one can distinguish between political, economic, and social cultures, each being a repository for values and norms that permit political, economic, and social strategies and behaviours to develop. Irina [164] comments on these cultural classifications, noting that political culture determines the political behaviours that are possible for whatever political persuasion is common, and this may, for instance, relate to political interactive alinements. Similarly, economic culture determines what economic behaviours are possible under, for instance, neoliberalism, especially when connected to free trade agreements. Finally, agency culture relates to attitudes towards civil society, for example, attitudes concerning human rights. There is an indication, however, that some ASEAN members are developing economically [165]. This may be the result of ASEAN's coordinated interventions as opposed to individual regional states adopting similar but uncoordinated developmental strategies. This latter possibility is more likely, since ASEAN has a lack of identity [166] that would be necessary for such coordination, and this would support the view that ASEAN has both a weak and a passive culture.

If it is the case that passive cultures exist within ASEAN, then this confronts the *Huntington paradox* [167] of political development—a tendency of political institutions to decay and become less effective over time as societies complexify. Here, policy innovations are encouraged by a distribution of power, which is neither highly concentrated nor widely spread. A study of the literature on innovation in organisations indicates that systems in which power is dispersed have many proposals for innovative reforms but few adoptions, and vice versa ([167], p. 85). In support of this, Jreaisat [24] suggests that what is needed in the development process is not the dispersion of power but its centralisation. We shall return to this later.

The distinction between a developmental agency and one that is purely instrumental is that the former is capable of learning while the latter is not, simply responding to its environment through a selected option in its existing repertoire of possible behaviours,

whether appropriate or not [168]. All cultural agencies have their own active or passive cultures, with the latter characterising ASEAN. If organisations are devoid of an active culture, it is problematic to create commonly shared values. Thus, in the case of ASEAN, it has an announced set of values (which include cultural pluralism, peace and security, cultural understanding, prosperity, noninterference, consultation/dialogue, adherence to international law and rules of trade, democracy, freedom, promotion and protection of human rights, and fostering a common voice in tackling extremism, lack of tolerance and respect for life, as well as social disharmony and distrust (Mun, 2019)). However, as already noted, none of these values has achieved a pragmatic outcome, indicating a stalled organisation, just as with such other attributes as human rights issues, democracy development, and equal distribution of income [27,90,95,128,144].

### 3. ASEAN and the Tönnies–Triandis Cognition Connection

To understand the behaviour and mindset of ASEAN, we used agency theory together with Tönnies social organisation theory, which was created in part 1 of this paper [1]. While there, we explored theory creation, here we demonstrate its application to MAT to improve its relevance for various agencies. MAT offers a thorough and all-encompassing perspective on agency, highlighting attributes that can have both positive and negative effects. By incorporating Tönnies' theory of social organisation into MAT, we introduce a novel foundational characteristic that reflects the operational stability of agency in culturally diverse and complex organisations.

In this section, we explain the Tönnies social organisation and Triandis theory of interpersonal behaviour as they relate to social organisation and a capacity to create collective action. We integrated Tönnies' theory of social organisation into MAT in part 1 and introduced a new formative trait indicative of agency operative stability in culturally diverse complex organisations. As a method of analysis, MAT is adaptable to various contexts, allowing for particular interventions to improve agency performance. Here, we explain the traits and how the Gemeinschaft–Gesellschaft paradigm and trait theory (MAT) can explain the ASEAN obstacles and pathologies to create favourite outcomes and what characteristics are dominant. The key concepts of MAT and Tönnies' ideas are elucidated in terms of dominant traits such as Patternism–Dramatism and Gemeinschaft–Gesellschaft. Gemeinschaft and Gesellschaft are sociological concepts that delineate different types of agency based on their interpersonal (social) relationships, with various influences shaping sociocognitive organisation derived from other traits within MAT relevant to context. Patterning and Dramatising are concepts originating from cultural anthropology [1]. In the previous section of this paper, it was expounded that "These multiple contextual layers, including macro and micro levels, influence how agents interact with each other and the agency's behaviour, which can be understood through the lens of sociocognitive style" ([1], p. 27). These levels play a role in how agents engage with each other and how agencies behave, offering insights through the framework of sociocognitive style.

Patternism is distinct from Dramatising in the same way that Gemeinschaft is distinct from Gesellschaft. Patternism values are symmetry, pattern, balance, and the dynamics of social relationships, while in Dramatism the key values are goal formation for individual benefit, and self-centred agencies operate through social contracts between individual members. The performance of organisations under the influence of culture showed that the performance of Dramatisers was significantly more successful than that of Patterners [169,170], which is similar to Asplund's finding in his survey, which explains the fact that while Gemeinschaft creates a location for productive work, Gesellschaft does not produce any utilities at all [90]. Also, here we need to note that Patternism is closely connected to Harmony and that a high level of Harmony can affect negatively creativity [171], while an excessive Harmony orientation may abolish all incentives to do anything [77].

There are similarities between Gemeinschaft and Patterning because both are associated with agencies that have strong bonds of solidarity, loyalty, and trust among agents. They tend to have a low degree of differentiation, complexity, and conflict among them,

and a high degree of stability, continuity, and harmony in their interactions. Gesellschaft and Dramatising are both associated with agencies having weak bonds of solidarity, loyalty, and trust among agents. Due to their characterisation of being associated with impersonal and formal relationships, they also tend to have a high degree of differentiation, complexity, and conflict, with a low degree of stability, continuity, and harmony in their interactions ([1], p. 27). As explained in part 1, there is a connection between Dramatising social orientation and Sensate cultural orientation, while Patterning social orientation is likely to be more connected with Ideational cultural orientation ([169], p. 16).

With this, Bosteels [84] refers to a term that describes the strengthening of the state's central power, where it prioritises the interests of private and transnational companies to serve its own benefits rather than collective benefits.

## 4. Discussion

This paper has argued that an effective way to analyse complex adaptive systems such as ROs is through the use of a metacybernetics approach and MAT as a schema to explore the hidden aspects of these systems. MAT provides insights into how an agency's culture and social structures influence its agents and shape its identity, allowing the identification of strengths and weaknesses that can ultimately improve its performance in achieving its mission. To explore social relationships between agents within complex agencies, the Tönnies social organisation paradigm is configured into MAT. The configuration-based approach provides a more comprehensive analysis of complex agencies, identifying specific issues that impact functionality, including stability and coherence. Multiple Identity Theory (MIT) and MAT are introduced from part 1 of the paper to address organisational paradoxes and contradictory tensions and provide an empirical setting for trait evaluations that can help evaluate RO stability and coherence. Having developed this theory, it should be noted that, methodologically, the approach could be enhanced. Such enhancements might include recognising that diagnosis may be part of a larger process of assessment, evaluation, intervention, or planning that aims to address the pathology and its effects. Thus, diagnosis can move to prognosis—an anticipation concerning likely outcomes given possible courses of action to address pathological conditions. This likely includes recommendations for pathology resolution, treatment, or management. Additionally, conducting a quantitative evaluation of ASEAN would be useful, and this would corroborate the qualitative evaluations we have developed. Moreover, an investigation of MAT could be undertaken with the aim of identifying the stability of the different mindsets indicated.

The paper applies this framework to ASEAN to diagnose its incoherence in terms nonfunctionality. It presents an effective framework for understanding the complexities of ROs, with a focus on how cultural agency, mindset agency, and social organisation interact to impact functionality. The MAT schema offers a useful tool for exploring how the collective mindsets of ASEAN shape its political culture and the capacity of its members to act together in regional affairs. The approach taken revisits the theoretical underpinnings of MAT as applied to ASEAN, recognising it as a complex system grounded in principles of cybernetics and critical realism. ASEAN possesses emergent properties that arise from the interaction amongst diverse entities within the system, requiring an in-depth understanding of underlying processes, systems, and agents. By applying the concept of MAT, the paper explores how the collective mindsets of ASEAN shape its political culture, which in turn shapes the organisation's capacity to act as a functional system.

To improve ASEAN's functionality, the paper recommends a shift towards a more autonomous, region-centric, and assertive sociopolitical–cultural orientation. ASEAN has a substructural cultural agency characterised by a collectivist sociocultural orientation and a strategic personality dominated by Incoherent Hierarchical Collectivism. The paper identifies the process intelligences of ASEAN, including its abilities to access and apply knowledge, self-organise, and create appropriate behaviour relative to contexts. ASEAN's passive culture and beliefs about authority inhibit self-sustaining responses to significant environmental situations.

We have examined ASEAN since it stands out through its inherent conflicts and paradoxes. We began by enhancing MAT to enable it to explore sociopolitical relationships by creating a configuration for the Tönnies paradigm of social organisation. It was argued that, on its own, this paradigm is inadequate to characterise an RO. In contrast, MAT is concerned with complex adaptive systems and investigates agencies in terms of a substructure, with dynamic variables constituted as formative intangible traits the values of which create structural imperatives. One of the traits proposed here, the sociocognitive style, derives from social relationships and cognitive style. While the former trait can take bipolar trait values of Gemeinschafts–Gesellschafts, the latter has bipolar values of Patterning–Dramatising. However, the sociocognitive style has bipolar values of coherence–incoherence and is capable of indicating the stability of an RO where its values could be determined.

Hidden substructure influences structure through formative traits. These are described by MAT with its five parameters, three of which relate to personality and two of which are sociocultural through the original cognitive style traits of Patterning–Dramatising and Ideational–Sensate cultural trait values. The sociocognitive style trait arises from the degree of commonality between the trait values of Gemeinschafts–Gesellschafts and the mindset traits values of Patternism–Dramatising in the same way as there is some commonality between Gemeinschafts–Gesellschafts and Triandis' trait values of Collectivism–Individualism. The Tönnies sociocognitive organisation trait is intimately connected with the operative system trait of mindset theory, and both theories may be envisaged to engage with interactive bipolar trait values. MAT explains how the four metaphenomenal traits can influence the social relationship trait with values of the cognitive style trait, and it delivers the sociocognitive style trait which may take values of coherence or incoherence. We recall that coherence means the degree of unity and coordination among ASEAN agents on regional issues. The ASEAN way is central to determining the mindset of ASEAN, recognising its principles and norms that guide the interactions between ASEAN agents. It implies an emphasis on the norms of consensus, sovereignty, and noninterference in the internal affairs of those agents and adopts fundamental and fully practised principles of noncoercion, which are central to the ASEAN way, and which means that the ASEAN agents do not use force or threats to influence or interfere with each other's internal affairs. The concept is related to the respect for sovereignty and noninterference, which are also part of the ASEAN way. The idea of noncoercion is that it can be used to promote peaceful and cooperative relations among the ASEAN agents and to avoid conflicts or disputes that may harm regional stability and security. All negotiations involving ASEAN agents embrace the principles of the ASEAN way, underlining the determined personality mindset that has been postulated for ASEAN.

The notion of coherence promoted by the ASEAN way is intended to: foster a sense of community and identity based on shared values and interests; enhance horizontal coherence among the various ASEAN-led mechanisms and initiatives, like the ASEAN Community and the ASEAN Regional Forum; and maintain vertical coherence between ASEAN and its external partners by promoting dialogue and cooperation on regional and global issues. However, while coherence is an expressed desire, it is not natural to ASEAN agents, since there are various factors that tend to create internal heterogeneity and tensions between them. For example, its organisational expansion in the 1990s was accompanied by a growing internal heterogeneity, resulting in internal tensions that the ASEAN way has not adequately addressed. Some of these tensions arise from different levels of economic development, political systems, security interests, and historical grievances among the agents. ASEAN coherence is also challenged by external pressures and influences from major powers, such as China and the US, with divergent or conflicting interests and agendas in the region. Economic development and trade are intricately intertwined with security concerns in the region, leading to tensions with countries such as Cambodia, which heavily relies on economic and financial support from China. This dependency has the potential to spark regional tensions within ASEAN, particularly regarding issues like the South China

Sea dispute, which can sow discord and create inconsistencies among member states. In this context, Cambodia's strong allegiance to China can sometimes outweigh its commitment to ASEAN's collective interests. Thus, coherence is an ideal of ASEAN that remains on its wish list. This supports the realisation that its Incoherent Hierarchical Collectivism mindset is inherently unstable, so that it is incapable of recognising or delivering the requisite pragmatic outputs to maintain itself and increase its level of viability.

Looking at ASEAN technically, through our theoretical lens, it is an agency with an operative Dramatist–Ideational sociocultural orientation, a personality mindset of Hierarchical Collectivism, and an agency mindset of Incoherent Hierarchical Collectivism. The cognition personality is defined in terms of Embeddedness, Harmony, and Hierarchy. The Ideational cultural attribute sees reality as supersensory. The strategic personality determines how the ASEAN culture understands and responds to reality. The ASEAN mindset involves Embeddedness, where values like social order, respect for tradition, security, and wisdom are especially important. The status quo is important, as are restraining actions or inclinations that might disrupt in-group solidarity or the traditional order. That these things may be important does not necessarily mean that they function well. This is due to the interactive interference of the different traits. Its affect mindset is Defensive Choleric, which has affect personality traits of Containment, Protection, and Dominance. Its sociocultural agency traits take Missionary and Empathetic values, the former imposing perspectives on others, the latter being responsive to others. Its Protection trait value is manifested through its attitude, characterised by measures of liberalisation (intended to improve the situation for agent investments), facilitation (to ease administrative needs concerning fiscal and business matters), promotion (through support by information flows and facilitation agencies), and regulation (to enable an improved fiscal environmental).

Collectivism–Individualism mindset outcomes are such that the Collectivism orientation tends to drive relational behaviour, with a tendency towards cooperative and harmonious orientations. The Individualism orientation tends to create more self-reliance, encouraging competitive behaviour. Balances may occur between these traits. MAT is a formative trait psychology bedded in a substructure that explains how mindsets (patters of affect, cognition, and behaviour) are formed and changed by interaction between agents and contexts. It is related to the paradigms of both Tönnies and Triandis. Mindset Agency Theory can be applied to different levels of analysis, such as individuals, groups, organisations, societies, and cultures. It can explain how different types of mindsets interact and influence each other across different levels and contexts. The Tönnies and Triandis paradigms are linked with MAT by recognising that they are complementary and interrelated. They can be seen as different dimensions of formative traits that can be applied in related forms of analysis. The three paradigms can explain how different types of formative traits interact and influence each other across different levels and contexts, though the Mindset Agency Theory is overarching. This means that it can incorporate the insights from the Tönnies and Triandis paradigms into a more comprehensive framework that accounts for the complexity and diversity of social systems and their personalities.

So, we have provided a more comprehensive theory related to the sociocognitive organisation through MAT that, as a metaphenomenal theory, can connect with both tangible and intangible variables, and which has the potential for an improved RO analysis that can provide behavioural predictions for determinable contexts. This approach enables a substructural understanding of ASEAN that focuses on different kinds of intelligence and that can explain ASEAN outcomes and the efficacy or inefficacy of ASEAN. ASEAN often boasts that it is the most successful organisation in Asia since its founding in 1967. However, its success is questionable, as its functionality as an RO has been poor. ASEAN has shown longevity, expansion, resilience, and influence over the past 50 years, despite various challenges. It has grown from 5 to 10 members, covering most of Southeast Asia. ASEAN claims that it is adaptive and flexible in responding to changing regional and international situations. It has become a key actor and platform for dialogue and cooperation in Asia and beyond, involving major powers such as China, Japan, India, Australia, and the

United States. However, these achievements are overshadowed by ASEAN's operational inefficiency, which makes it a weak and ineffective organisation that has failed to deliver pragmatic outputs in its regional affairs. Its performance and credibility are hampered by: (a) its shortcomings and limitations, such as its lack of political will, institutional capacity, and enforcement mechanisms to implement or ensure compliance with its agreements or decisions (e.g., it was unable to help resolve the South China Sea disputes, protect the Rohingya minority, restore democracy in Myanmar, or contain the COVID-19 pandemic); (b) its internal divisions, divergent interests, and external pressures that undermine its cohesion and centrality (e.g., it was unable to present a unified stance or response to China's growing influence and assertiveness, US strategic rivalry and withdrawal, or the Indo-Pacific concept and strategy); and (c) its failure to adapt to changing regional and international environments and to meet the expectations and needs of its people and partners (e.g., it has been unable to address the challenges of digitalisation, innovation, and sustainability or to promote human rights, democracy, and civil society).

ASEAN has a forum in which its institutional norms and rules (like the ASEAN Way or ASEAN Centrality) operate. By this means, ASEAN draws diplomatic attention from great powers, and since it is a 10-member-state regional organisation that can (at times) speak with one voice, great powers find it attractive because if the member states support what they are doing, their actions take on "legitimised" labels from Southeast Asia. External powers, then, support ASEAN diplomatically and financially, and, even though the Secretariat is small, it functions well. This means that ASEAN depends on attention from the great powers, and if they ignore these powers, then the organisation will be weakened, and its dependency on external influences/forces will limit its self-reliability. So, ASEAN development and processes are related to external forces and players, and these are therefore able to direct ASEAN, putting into question its degree of autonomy.

ASEAN's own basic principles seem to have been an obstacle to closer integration between its member states, examples being harmony, noninterference, and a consensus-based decision-making process with a decentralised structure. All of these principles alone can weaken integration and cooperation in any organisation. Harmony organisation is idea-centred rather than problem-solving-centred, so ASEAN already has a predisposition to not be very pragmatic. This is a different condition from the serious pathology that arises when agency instabilities arise, disabling requisite adaptive strategies to change. ASEAN can create proposals with little capacity for adaptation and implementation, and in its decentralised system power is widely distributed. Following Huntington's proposition that systems in which power is concentrated have few reform proposals but many adoptions, there is an argument that for development towards improvement, centralisation is better than decentralisation, which has the potential to create a burden across the population of agents by exacerbating such facets as knowledge deficits, goal conflicts, and miscommunication. A third, distributed option is possible, which is a decentralised system that has no central authority but consists of many independent and equal nodes that cooperate and communicate with each other. This can be more resilient and democratic than the alternatives that are beyond any single point of failure or control, though it can suffer from challenges concerning communication, coordination, security, and performance. ASEAN is challenged in all of these areas.

The institutions of ASEAN have a relatively low level of development regarding improvement as defined by their mission, and this is because of the state-centric approach to cooperation. This results in national interests being of greater importance than ASEAN common interest, where national state sovereignty is unquestioned. ASEAN's autonomy has not increased significantly, it has not made any major institutional innovations, and no objective functional demand arises from any specific interactions between member states. We have already noted the comment by Jones and Smith that goes even further, indicating that ASEAN is making process rather than progress, and it can only offer a platform of limited intergovernmental and bureaucratically rigid interaction. Decision making is based on consensus, making it difficult to reach conclusions, and this often results in policy

detail being delivered later, at some unreachable temporal horizon. ASEAN's legal base also affects obstacles that inhibit the creation of positive outcomes, and the lack of an independent entity character is one of the principal reasons why ASEAN is slow, not only in reaching agreements, but also in implementing them ([172], p. 18). ASEAN operates on the principle of static rather than dynamic development and stability, upholding existing conditions, resisting change in the region, and maintaining the current balance of power and interests among its member states and external actors to safeguard its unity and central role.

This is not to say that ASEAN does not implement agreements, and here an example might be useful [173–176]. It established two centres to implement the ASEAN Agreement on Disaster Management and Emergency Response (AADMER), which is a legally binding regional policy framework for disaster risk reduction and management and intended to primarily act as a monitor for ASEAN. The humanitarian assistance centre and the coordinating centre for humanitarian assistance were set up in 2011 to facilitate and coordinate the delivery of humanitarian aid and disaster relief in the ASEAN region. These centres played a role in the 2017 crisis in Bangladesh and Myanmar, where the Rohingya people faced persecution and violence perpetrated by the Myanmar military government. The centres provided rice, personal protective equipment, medical supplies, and food items to Rakhine State for the Rohingya refugees and asylum seekers. However, ASEAN's support for the crisis was insufficient and ineffective. ASEAN only issued statements that expressed concern but did not propose any concrete actions. It also sought dialogue to create trust and understanding between actors but without any tangible outcomes. Moreover, it proposed a five-point consensus plan that was unclear, voluntary, and lacked a timeline for implementation. A more effective approach would have been to apply sanctions to the Myanmar military government, as some countries outside ASEAN have done, to create negative consequences and incentives for them to stop their repression and violence. This would only work within ASEAN if there were a regulatory framework that could control the benefits that its agents receive from being part of the regional bloc. Such a loss would have to be more substantial than the huge reduction in trade that Myanmar has experienced (caused by a spontaneous response to protests over its violent behaviour). However, unlike the EU, ASEAN does not have such a framework and has been unable to create pressure to resolve the Myanmar conflict. In particular, while ASEAN has provided some humanitarian assistance to the Rohingya crisis, it has failed to address the root causes of the conflict or to hold the Myanmar military government accountable.

To avoid conflicts with its member agents, ASEAN adopts a wide frame of reference that is intended to take into account multiple attributes, perspectives, values, and interests. In principle, this should enable issues to be classified, where each classification has a general regulatory response that, with reflexive analysis, might be considered appropriate for conflict resolution. This would require specific local contexts to be explored in sufficient detail, enabling a set of rules to be created for local ASEAN action. However, this does not occur, since, as we have argued, ASEAN does not delve into the details of given situations. A wide frame of reference seeks a balance between responding to specific issues and maintaining regional peace and stability, thereby, it is claimed, allowing an adaptive and evolving approach to changing circumstances and needs. It also enables ASEAN to claim that it respects the sovereignty and autonomy of its membership by not intervening. A further claim is that this enables a dynamic and flexible response to situations. However, any such responses are meaningless since ASEAN does not intervene, and its lack of pragmatism means that it avoids action for specific issues. In place of this, ASEAN creates agreements that are dependent on the ad hoc voluntary compliance of member agents without the anchor of a common political culture. This is illustrated by the realisation that ASEAN declarations and statements commonly adopt the word "shall", and this refers to intention. This highlights that, despite conditional wording, definitions and statements are devoid of meaning, especially concerning undefined terms like democracy, human rights, and integral economic development. ASEAN has not even been able to

resolve regional tensions between member countries or respond to intraregional or regional military conflicts by issuing common statements or adopting common policies/politics, for example, in the current South China Sea conflicts. The South China Sea issue creates both regional tension and geopolitical pressure for ASEAN and its member states. With the absence of definitions and a lack of a measurement system, it has no means of measuring outcomes against intentions. All of this taken together makes ASEAN integration rather shallow, its conditional statements leading to proportional integration, which means the statements are made without a plan and real aspiration for implementation. Processes of integration and an increased level of cooperation occur mainly on paper but not in practice, and they are devoid of a legal basis. Proportional integration has led to poor performance. Such factors are normally adopted to measure degrees of regional integration. The level of integration it has managed, as well as ASEAN's performance concerning democracy and human rights, are seen to be regressive, and its level of economic cooperation has been shown not to have significantly developed during the last 25 years concerning intra-trade or intra-Foreign Direct Investment.

The proposition has been offered that ASEAN's development as an operatively efficacious organisation is only feasible if it can maintain a personality driven by a coherent political culture that is neither weak nor passive. Here, political culture orients the agent macroscopically, influencing its personality and potential for behaviour. We can explain the potential for a declining, increasing, or stationary RO development in cybernetic terms. While declining or increasing development is dependent on the cultural orientation of an RO, stationary development (or nondevelopment) occurs when the culture is incapable of change. While figurative intelligences can be used for Ideational creativity, its pragmatic capacity is not supported and it may therefore suffer from learning inefficacy in this respect. This appears to be the case with ASEAN. Its member agents have all the factors that can establish it as a global-level player and an actor in international as well as regional affairs. It has a young population, a strong production base, a high number of foreign currencies in central banks, and fast economic growth. All these factors should create a strong and coherent platform for ASEAN cooperation. However, its member agents must increase their level of collective action, and it seems that the traditional ideas for a "collective ASEAN" that its agents still adhere to mean that it is unable to create state-level collective actions.

To enable ASEAN to overcome its stagnation (if not decline), it requires a language shift, using "must" rather than "should" or "shall" and thus moving away from a weak political culture and identifying its figurative intelligence pathologies, enabling it to maintain an active and pragmatic political culture, which requires a degree of shift towards a Sensate cultural trait. This would enable it to develop a paradigm that enables it to operate coherently and an orientation that satisfies its potential for efficaciousness, moving from a state-centric approach to a region-centric one. This shift would depend on the ruling elites at the state level and their willingness to support development and share power. Also, ASEAN member states must question the harmony organisation with a consensus-oriented decision-making process. When consensus is the priority over ASEAN efficacy, ASEAN will hardly be able to achieve any of its desires. ASEAN agents need to become more autonomous so that their behaviour can shift from being that of an instrumental organisation, thereby enabling it to be less dependent in its functionality on arbitrary environmental events.

RO relationships need to be such that collective action is feasible, and mindsets enable the potential for levels of cooperation and collective action. A collective agency operates through shared beliefs, pooled understanding, group aspirations, incentive systems, collective action, and efficacious processes and behaviours associated with particular mindsets. Collective action refers to action taken together (collectively) based on a collective decision by a group of people whose goal is to enhance their condition and achieve a common objective. The traits that underpin mindsets derive from the dominant values in a society. People and organisations with these values are therefore likely to do better in that society than those who have different values. Combinations of traits, expressed in terms of bipolar value pairs, are determinants of behaviour, though it must be realised that the traits

can mutually influence each other. It cannot be assumed that some traits are "better" or more effective than others; they just create the tendential ambient characteristics indicative of individuals, organisations, or states, thus providing tools to predict how they might respond to given situations in given contexts. Eight different types of cognition mindset (Collectivism–Individualism) and affect mindset (Stimulation–Containment) have been identified. The types of these mindsets are trait-dependent and guide how agents may interact together, and that interaction can in turn influence the agency mindset.

A synopsis of ASEAN can be provided as follows. It exhibits a Collectivist approach characterised by a blend of a Dramatist–Ideational sociocultural orientation and a strategic demeanour primarily shaped by Incoherent Hierarchical Collectivism, which is characterised by concepts of Embeddedness, Harmony, and Hierarchy. In terms of its social orientation as an entity, ASEAN displays a Patterning trait, where the arrangement of relationships holds significance, indicating the varying positions individuals and groups hold concerning one another, thus influencing the societal structure. Symmetry, patterns, balance, and the dynamics of relationships play a crucial role, suggesting the presence of a trust-building aspect. ASEAN has a passive culture that is hardly capable of applying cultural knowledge or learning or creativity, such constrictions being due to its beliefs about authority, inhibiting self-sustaining responses to significant environmental situations. It has an orientation that supports the Ideational, for which reality is seen as supersensory, and where the consequences of the psyche and thought are significant, morality is unconditional, and tradition (nationality) is of importance. While ASEAN tends to rely on personal relationships cemented by trust in their ingroups, they are more careful with outgroups, implying that ingroup collective action is much easier to create than outgroup collective action, for which there is little process of socialisation.

ASEAN agents have different orientations and preferences and different ways of thinking, feeling, and acting when they communicate and relate to each other and when they cooperate and coordinate with each other. These different ways are influenced by their cultures, which are the shared values, norms, beliefs, and practices that shape their collective identities and behaviours. According to some studies, ASEAN agents tend to have a Collectivistic culture, which means that they value group harmony, loyalty, and solidarity over individual autonomy, rights, and interests. They are also said to have a high-context communication style, which means that they rely more on implicit cues, nonverbal signals, and personal relationships, rather than on explicit words, verbal messages, and formal rules. Moreover, they tend to have an ingroup–outgroup distinction, which means that they differentiate between people who belong to their group (ingroup) and people who belong to other groups (outgroup). As a result, ASEAN members tend to rely on personal relationships cemented by trust in their ingroups, while they are more careful with outgroups. This implies that ingroup collective action is much easier to create than outgroup collective action for ASEAN members. For outgroup collective action to be created, there needs to be more socialisation, which means more interaction, communication, and exchange of information among outgroup members to build trust and understanding.

ASEAN has a weak, passive, and loose culture that lacks strong influences or values. It also follows a principle of nonintervention, which means that it does not interfere with the internal affairs of its member states or other countries. This makes it appear to be an illusory rather than a real organisation. ASEAN's process intelligences, which are its abilities to access and apply knowledge, to self-organise, and to create appropriate behaviour relative to contexts, are not effective. Its agency function, through an ability to manifest its mission and goals, pragmatically indicates both organisational instability and inconsistency [177]. It seems to be declining rather than improving as it faces increasing complexity and challenges from its environment, including conflicts, disasters, and globalisation. To survive, it has made some adjustments, such as adopting more formal meetings instead of informal ones. Formal meetings have more structure, preparation, and documentation than informal

ones. However, this is only a small change that does not make ASEAN more pragmatic or proactive.

The MAT model and its derivative mindsets have been used to illustrate that ROs, as cultural agencies, always have the potential to be dynamic, adaptive, self-organising, proactive, self-regulating sociocognitive and socioaffective autonomous plural agencies. They interact with their social environments, and from these they acquire intrinsic information [178,179]. This can be defined as the information that is inherent to a complex and uncertain structure or process that reflects its essential nature or character and is valuable for decision processes regardless. It can be contrasted with extrinsic information, which is information that is derived from or influenced by external sources, such as observations, feedback, models, or expectations. Intrinsic information enables agencies to maintain their stability, unless they are subject to inherent pathological conditions, as in the case of ASEAN.

It would seem, for instance, that the cognition agency mindset dominated by Incoherent Hierarchical Collectivism we have assigned to ASEAN is not inherently stable, though the Hierarchical Collectivism is likely stable. Maruyama's [180] inquiries originally identified four stable mindscapes that have meaning equivalence to four personality mindsets [2], and so in investigations of the stability of mindsets, some attention might be allocated there. ASEAN personality can be seen as a normative set of logical mental rules and strategies, while the collective mind is seen as an information system that operates through a normative set of logical mental rules and strategies [181–184]. These rules and strategies may fail when pathologies develop, either through internal or external forces. Regarding the main traits and values of ASEAN, there are five formative traits, three of which define dispositional personality and one each for cultural orientation and social orientation (interaction with the social environment).

## 5. Conclusions

In this study of ASEAN, we have considered the RO through the veil of MAT as a qualitative indicative study of the organisation. This has been carried out by linking appropriate mindset values to opinions and evidence found in the literature. The most important attribute is that of cultural instability due to its capacity for sustentation. Cultural stability arises when conflicting values within ASEAN member states lead to disruption in the coherence of the system. This can act as a catalyst for other instabilities and subsequent pathologies. Instability, defined as a condition that disrupts agency coherence, renders the system more vulnerable to internal or external perturbations, ultimately resulting in pathologies—dysfunctions or maladaptations that negatively impact the system's performance or outcomes. The specific cause of instability determines the nature of the ensuing pathology.

Cultural instability for any RO occurs when the values that construct it conflict, and this can impact its identity and purposes. Five sources of cultural instability have been identified that can trigger different pathologies and consequences for ASEAN. In Table 2 we summarise the sources, pathologies, and consequences, selected because they represent the most salient and pressing issues that ASEAN faces in the current and future regional and global environment. They also reflect the diversity and complexity of ASEAN's historical, political, economic, and social contexts, as well as the opportunities and challenges that they entail.

A conceptual alignment between the observed behavioural paradox and narcissism can be established, given that both stem from a discrepancy between the projected self-image and the actions of the organisation [185]. An RO possesses a dispositional operative affect trait for emotion management, characterised by emotional properties like control, domination, a quest for supremacy, hegemony, power seeking, situational pre-eminence, sovereignty, ascendancy, authority, command over dominion, and susceptibility to narcissism and vanity [1]. ROs susceptible to narcissism may assume vulnerable/covert or grandiose/overt properties [186], with other forms also identifiable [187,188]. Covert

narcissists exhibit emotional traits such as low self-esteem, insecurity, hypersensitivity, defensiveness, and shame, often rooted in pathological conditions from early-life abuse or trauma. They project Collectivism through external unity and collaboration while internally grappling with the diversity, inequality, and conflicts that resonate with Individualism. This can result in the behavioural paradox driven by the Collectivism–Individualism contradiction. In contrast, overt narcissists possess characteristics like high self-esteem, self-confidence, and a desire for admiration, leading to consistent self-presentation and positive feedback. They are less likely to exhibit a behavioural paradox as their external image aligns with internal self-perception, avoiding contradictions between projected self-image and observed actions.

**Table 2.** Consequences of ASEAN Instabilities.

| Source of Instability | Pathology | Consequence |
|---|---|---|
| Cultural Heterogeneity | Nihilism | Loss of purpose and direction, diminishing ASEAN's legitimacy and effectiveness |
| Developmental Inequality | Decadence | Human rights abuses and corruption, negatively impacting ASEAN's credibility, reputation, and contributions to the global common good |
| Geopolitical Pressure | Violence | Aggressive and hostile behaviour, potentially leading to the use of force and coercion, affecting ASEAN's peace, stability, and accountability to the international community |
| Regional Tension | Fanaticism | Intolerance and oppression of minorities and vulnerable groups, negatively impacting ASEAN's diversity, harmony, and respect for democracy and human rights |
| Identity Schism | Narcissism | Self-centred and arrogant behaviour, leading to the exploitation of others' needs and interests, resulting in paradoxical behaviour, reducing ASEAN's consistency, reliability, and trust |

This explanation can be applied to the complex adaptive system that is ASEAN, defined through the plurality of its population of agents. Embodying covert narcissism, its behavioural paradox is demonstrated when its external facade contrasts sharply with internal challenges, impacting its ability to develop and integrate. This is illustrated by the many identifiable events that include: missing common identity which may create pathologies to create ASEAN policy and culture as inefficacy of collective actions. Anti-communism once gave ASEAN a common identity. However, with the collapse of communism and the end of the Cold War, that identity became redundant, and ASEAN now struggles to create a new common identity. Despite the world's transition from a bipolar to a multipolar paradigm, ASEAN encounters difficulties in finding its own identity and position globally, hindering its ability to act as a global player on the international stage, despite its size and potential capacity.

Every decade, ASEAN has faced difficulties in responding to crises related to Asian or ASEAN affairs, such as the coup in Cambodia just before it was due to join ASEAN and the "fog crisis" in the maritime areas of Southeast Asia caused by huge forest fires in various parts of Indonesia or floods in South Thailand [19]. ASEAN was unable to assist with these events. For example, during the Haiyan Typhoon in the Philippines, ASEAN was only able to establish an observation group, while the European Union and the US Navy were able to assist the Philippines. ASEAN typically does not respond collectively to regional crises when they occur, but instead is more inclined to formulate new ASEAN mechanisms after a crisis has passed, as seen with the Asian financial crises of 1997 and the creation of the CMI after 2000 [95]. Similarly, during the COVID-19 pandemic, ASEAN failed to establish a common framework for regional health responses, hindering a coherent pandemic response. A similar finding was observed in the SEAS report, which found that 49% of respondents (elites) believed that ASEAN was unable to recover from the pandemic [40]. There is more evidence of failures than successes. Shortly after the coup in Cambodia, the East Timor crisis occurred, and ASEAN was unable to effectively address it, with external agencies

such as the IMF and the UN becoming involved instead. Later, similar issues arose, such as during the Haiyan typhoon in the Philippines in 2013, where ASEAN's response was sluggish, and during the search for the missing Malaysian aeroplane MH370, where rescue operations were conducted by Australia rather than ASEAN. Finally, in the humanitarian Rohingya crisis in Myanmar, ASEAN has struggled to take effective action. Also, we cannot forget ASEAN's action and response to South China issues and the never-ending story of the CoC, which is still in the process after many decades [27,89].

The manifestation of covert narcissism is evident in ASEAN's attempt to present a united front to the international community as it seeks to navigate its internal complexities, exposing the profound disparity between its external image and internal reality. This can be traced back to its early development in response to external threats such as communism, colonialism, and hegemonic powers. Established to maintain regional peace and stability, ASEAN encountered internal challenges due to its cultural social and economic diversity. To address these challenges, ASEAN adopted its noninterference principle, reflecting low self-esteem and insecurity as it aimed to prevent the loss of any of the agents in its population or succumbing to external dominance. The consensus-based decision-making process further demonstrated defensiveness and paradoxical behaviour striving to preserve harmony and unity but impeding efficiency. This analysis, which recognises the fundamental Collectivism–Individualism conflict, responds to Hofstede's [189] analytic misstep which has resulted in the popular view that the Eastern part of the world is essentially Collectivistic and the West Individualistic, while in reality Eastern Collectivism is associated with family and familism, while in the West it is associated with organisations.

ASEAN's covert narcissism constitutes a formidable obstacle to its development. Addressing this challenge and mitigating behavioural paradox is important for cultivating a stable cultural condition. Such a positive transformation promises several outcomes, including a fortified sense of identity and solidarity, enhanced cooperation and integration, a profound respect for diversity and democracy, a steadfast commitment to peace and security, and a substantial contribution to the global common good. To understand how ASEAN can achieve this transformation, a CAT diagnosis offers a comprehensive and dynamic perspective on its behaviour and identity, while an MAT analysis explains its character.

CAT, a metacybernetic derivative, is a framework that studies complex adaptive systems such as ASEAN. Culture provides a self-stabilising (sustentative) mechanism through knowledge, shared values, beliefs, and norms that guide the behaviour and identity of an agency. However, ASEAN does not have an existential culture but rather an image of one defined by a set of artificial or imposed rules and regulations that constrain its autonomy and creativity. ASEAN's fictional culture is a result of its decision-making process, which is based on consensus, consultation, and noninterference among its member-state agents to avoid changes in stability and keep harmony. This process respects the sovereignty and diversity of each country, but it also limits the collective action and intervention potential for ASEAN.

Another challenge that ASEAN faces in its decision-making process is related to its lack of authority. Authority in this context is the ability to act or speak on behalf of agents without contradiction or spurious outcomes. ASEAN lacks authority, as it does not have a supranational body or institution that can enforce its decisions or impose sanctions on noncompliant agents. It can only act as a facilitator or coordinator for certain policy options or initiatives; it cannot impose its will or preferences on its agents. An element of this is that ASEAN can only speak on behalf of its agents when there is a consensus or a common position among them. Otherwise, it may face difficulties in representing diverse views or agent interests.

Now, ASEAN is an autopoietic agency in which its network of processes contributes to the creation and maintenance of agency identity through the self-production of its components and interactions. The processes enable adaptation and the ability of agencies to adjust to new information and experiences, this being essential for it to maintain its

viability. However, the network of processes may not be coherent and may not necessarily respond to requisite adaptive needs, for instance, by preserving existing agency structure and function, even where that structure and function is not beneficial to agency viability. The processes can also be purely self-referential, relying only on internal logic and supposed coherence. Such suppositions can be ignorant of any lack of correlation between actions that are responses to impactful environmental events and part of the ASEAN mission. This can result in autopoietic instability, for instance, leading to loss of connection or communication with its agents or other agencies.

To enhance the approach undertaken in this paper, further steps could be taken that consider how culture can facilitate self-sustainability as well as how the network of processes that are responsible for stability in self-producing processes can be addressed. Some possible steps to facilitate prognosis are to use a metacybernetic approach to understand the interrelationships and interdependencies among the cultural, strategic, and operative elements and processes, developing a learning culture that fosters innovation and feedback loops among ASEAN's members and stakeholders and balancing the autonomy and diversity of ASEAN's agents and institutions with the alignment and integration of ASEAN's vision and strategy.

Prognosis may arise in various ways. For instance, to develop an existential culture, the analysis suggests that ASEAN needs to reflect on the principles of metacybernetics and adopt certain steps. These include: (1) Identifying the core values and principles that define the identity and purpose of ASEAN. For instance, some of the core values and principles of ASEAN are peace, stability, cooperation, mutual respect, and diversity; (2) Developing a vision and a strategy that align with the core values and principles of ASEAN. For example, one of the visions of ASEAN is to become a community of caring societies, and one of the strategies of ASEAN is to enhance connectivity and integration among its members; (3) Implementing its vision and strategy through appropriate actions and mechanisms (via well-funded functional institutions) that reflect the core values and principles of ASEAN. For instance, some of the actions and mechanisms of ASEAN are indicated in the ASEAN Charter, the ASEAN Socio-Cultural Community, and the ASEAN Regional Forum; (4) Evaluating the outcomes and impacts of the actions and mechanisms using feedback loops and learning processes that enhance the core values and principles of ASEAN. For example, some of the outcomes and impacts of ASEAN are reductions in poverty, the promotion of human rights, and the prevention of conflicts.

ASEAN might well consider strengthening its role of authority in some areas of decision making, such as security, human rights, and trade. This could involve drawing on such considerations as (1) Revising its decision-making mode from consensus to majority voting or qualified majority voting, which can speed up the process and reduce the veto power of individual members. This could help ASEAN to address urgent or complex issues that require collective action or intervention more effectively; (2) Enhancing its legal framework by creating a binding dispute settlement mechanism or a court of justice, which could ensure the compliance and implementation of its decisions. This could help ASEAN to enforce its rules and regulations more consistently and credibly; (3) Recognising the institutional role in fostering regional integration and increasing its institutional capacity by establishing a permanent secretariat or a commission. This could coordinate and monitor the actions and mechanisms of ASEAN. It could also help ASEAN to improve its communication and cooperation more efficiently and transparently; (4) ASEAN should also realise the importance of leadership and the value of having a leading country. This requires that it should move from a state-centric approach to an ASEAN regional approach. For this to occur, ASEAN member states need to change their mindset, diminishing State Individualism that mistakenly enshrines national sovereignty and enhancing ASEAN Collectivism, a need to stand against its slow stagnant regional development that enshrines its role as a back-row player in not only the international arena, but even in the Asian regional arena.

To navigate and understand the intricate dynamics of ASEAN in its international environment, we have employed MAT to provide a comprehensive modelling, diagnosis, and analysis of ASEAN's complex nature. By delineating eight cognitive mindsets and eight affective mindsets, the paper delves into their nuanced interactions and mutual influences. Furthermore, it explores the adaptive capacity of ASEAN's mindset, showcasing how it can change based on context and analytical focus. This adaptability, in turn, has profound implications for the organisation's behaviour and overall performance. The multidimensional exploration of mindset agency thus provides a rich understanding of the factors shaping ASEAN's actions and responses.

This study and the diagnosis it generates could be validated quantitatively, and there are two ways of doing this. One way is to use trait questionnaires given to appropriately selected candidates within ASEAN, as explained by Yolles and Fink [2]. An alternative to this is to look for quantitative variables that can be used to directly support the current qualitative diagnosis. In Table 3, we suggest possible variables that could act as a basis for a quantitative inquiry to assess the functional coherence of ASEAN. These variables include ASEAN Authority, measured through an authority assessment to evaluate its influence over member states; Stability/Coherence, gauged by a stability index reflecting consistency in actions; Regional Infrastructure, assessed through an infrastructure investment index reflecting progress in connectivity; the ASEAN Secretariat Budget, analysed by historical budget allocation trends indicating shifts in priorities; Operative Paradoxes, measured through paradox quantification to identify and measure inconsistencies in ASEAN functioning; Collective Heterogeneity, assessed by a diversity index incorporating GDP, population, and cultural indicators; Intra-ASEAN Investment, determined by Foreign Direct Investment levels reflecting trust among member states; and Trade Volume, measured by a trade flow index indicating economic interdependence and collaboration within the ASEAN region. These variables collectively provide a comprehensive understanding of ASEAN's organisational dynamics and regional cooperation. However, accumulating these data will be for a future paper.

**Table 3.** Possible quantitative variables to be measured to support the analysis.

| Variable | Functional Coherence | Measurement Approach | Interpretation |
|---|---|---|---|
| ASEAN Authority | Level of authority ASEAN holds over its membership | Authority Assessment: Evaluate ASEAN's influence on member states' behaviour | This measure helps us understand the extent of ASEAN's control and impact on its member countries. |
| Stability/Coherence | Degree of cultural stability and operative coherence within ASEAN | Stability Index: Assess consistency and predictability in ASEAN actions and decisions Cultural distance: Differences in values and behaviours | A higher stability index indicates a more reliable and harmonious functioning of ASEAN as a regional organisation. Cultural values may be dominantly in conflict or harmony. |
| Regional Infrastructure | Improved connectivity and cooperation | Infrastructure Investment Index: Combine data on roads, ports, energy grids, and digital infrastructure | This measure reflects the progress in enhancing regional connectivity and collaboration through infrastructure. |
| ASEAN Secretariat Budget | Shifts in commitment and institutional support | Budget Trends: Analyse historical budget allocation trends for the ASEAN Secretariat | Changes in budget allocations reveal shifts in ASEAN's priorities and commitment to its institutional framework. |
| Operative Paradoxes | Contradictions or challenges faced by ASEAN | Paradox Quantification: Identify and measure inconsistencies in ASEAN functioning | Changes in paradox frequencies over time help identify the sustentative capacity, indicating a trajectory of improvement or not. |

**Table 3.** *Cont.*

| Variable | Functional Coherence | Measurement Approach | Interpretation |
|---|---|---|---|
| Collective Heterogeneity | Inherent diversity among ASEAN member states | Diversity Index: Incorporate GDP per capita, population size, and cultural indicators | The diversity index highlights the unique characteristics and challenges posed by different member states. |
| Intra-ASEAN Investment | Trust and confidence among member states | Foreign Direct Investment (FDI) within ASEAN | FDI levels indicate the level of trust and economic cooperation among ASEAN nations. |
| Trade Volume | Economic interdependence and cooperation | Trade Flow Index: Total intra-ASEAN trade value (exports + imports) | Higher trade volumes signify stronger economic ties and collaboration within the ASEAN region. |

## 6. A Caveat: Metaphor, Theory, and Configuration

In the realm of complexity theory, the strategic use of metaphors plays a crucial role in making abstract concepts more accessible by drawing parallels with familiar experiences. This practice is especially prominent in interdisciplinary fields grappling with complex systems, where metaphorical language serves as a bridge to understanding. However, it is important to recognise the limitations of metaphors. In contrast, theories offer comprehensive and coherent sets of principles, usually rooted in empirical evidence, that clarify and predict phenomena within specific domains. The validation of theories relies on rigorous testing and refinement processes, demanding tangible evidence of their predictive and explanatory power.

Metaphors serve as the seeds from which theories can grow [190], but they require substantiation through evidence and analysis to develop into fully realised conceptual frameworks. For instance, contingency theory, as adopted in part 1 of the paper, transcends its metaphorical origins to provide actionable insights across various theoretical domains. By moving beyond mere analogy, contingency theory emerges as a substantive framework, shedding light on the complexities inherent in systems.

While metaphors are essential for initial understanding and communication, theories must embody empirical rigour and practical applicability to establish themselves as foundational tools of scientific inquiry. Theoretical constructs go beyond metaphorical representation, evolving into robust frameworks shaped by empirical evidence and methodological rigour. The integration of stability theory, Von Foerster's insights, and Varela's autopoiesis [191] exemplifies this evolution, highlighting the dynamic nature of theoretical frameworks and their adaptation based on empirical observations. Exploring processes of configuration, MAT serves as a diagnostic tool intertwining trait instabilities with agency pathologies, grounded in empirical evidence and theoretical rigour rather than relying solely on metaphorical expression. These processes facilitate the transfer of concepts across different domains, using metaphors as cognitive devices to comprehend and depict complex abstract ideas.

By extending the elements that construct MAT from their psychological origins to the realm of agency theory and incorporating insights from Tönnies' sociological framework, we enrich our understanding of agency dynamics. Metaphorical strands that emphasise interactional patterns, hierarchical structures, and coordination paradigms bridge psychological, sociological, and agency-theoretical realms. Refined through rigorous analysis and empirical scrutiny, MAT emerges as a robust framework for inquiries into social and political complexity, offering subtle insights into agency behaviour.

Furthermore, in grappling with epistemological challenges and considering the complexity of complex systems, we must scrutinise the interplay between metaphor and theory. Acknowledging the cognitive depth inherent in metaphors and recognising the intricate dynamics of complexity allows for a more comprehensive understanding of their role in shaping theoretical frameworks. By navigating these complexities, we enrich our intellectual landscape and gain deeper insights into the intricacies of cognitive reasoning. This

involves not only recognising the metaphorical use of fractal patterns but also delving into the semantic and logical structures underpinning the concepts presented.

**Author Contributions:** Conceptualisation, T.R. and M.Y.; Formal Analysis, T.R. and M.Y. All authors have read and agreed to the published version of the manuscript.

**Funding:** This research received no external funding.

**Data Availability Statement:** Data are contained within the article.

**Conflicts of Interest:** The authors declare no conflict of interest.

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
