# Peer review of "Diagnosing Complex Organisations with Diverse Cultures—Part 2: Application to ASEAN"

_systems, doi:10.3390/systems12030107_

Round 1

Reviewer 1 Report

Comments and Suggestions for Authors

The document is well presented, however needs to be more reader friendly. You can for instance consider the use of systemic tools to visualize the conceptual; relations you are presenting in this theoretical document.

Following the previous observation, is not explicitly evident the link of MAT and cybernetics (metacybernetic framework) just to mention some of the core concepts and methods integrated in the document. 

The visualization of the model is important to make evident the predictive/anticipatory of the model proposed, as it list several variables subject to some form of measurement (e.g. Cultural distance), which adding to the CAS nature of the subject of study, offers the possiblity to validate the model by using techniques like Agent modelling.

Author Response

We thank you for your important contribution in improving this paper

Reviewer: Following the previous observation, is not explicitly evident the link of MAT and cybernetics (metacybernetic framework) just to mention some of the core concepts and methods integrated in the document.

Reply: Additional text and a new figure from part 1 has been introduced into the introduction to satisfy this very useful observation. We thank you for this advice.

Reviewer: The visualization of the model is important to make evident the predictive/anticipatory of the model proposed, as it list several variables subject to some form of measurement (e.g. Cultural distance), which adding to the CAS nature of the subject of study, offers the possibility to validate the model by using techniques like Agent modelling.

Reply: This is also very good advice. We have embedded this in a broader range of measures in an extension to the conclusion.

Reviewer 2 Report

Comments and Suggestions for Authors

The study represents the second part of a more complex study, regarding the cultural stability of ASEAN.

The analysis tool, developed on the background of social cybernetics, is Mindset Agency Theory (MAT) applied in a metacybernetic framework.

Unfortunately, the theoretical framework underlying the analysis is presented in the first part of the paper.

Considering the regional organizations (RO) as complex adaptive systems, the study explores the attributes that are the basis of ASEAN in order to determine its cultural stability and the possible ways of manifesting pathologies within ASEAN.

Moreover, even the fundamental research question of the current study focuses on solving the possible pathological problems arising from the cultural diversity and institutional dynamics of ASEAN and the regional organizations that belong to it,

Interfering with some aspects of a cybernetic nature presented in the first part of the study, the authors develop a specific gear with systemic and reflexive values governed by MAT.

Through MAT, the way in which the agency's behavior and performance are influenced by its mentality is elucidated.

Structured in several sections, the main characteristics of ASEAN and its behavior are addressed in the study, its operational coherence in the absence of which cultural instability appears within the various organizations.

Finally, implications and global analyzes are presented.

I noticed that, in the case of describing ASEAN, MAT offers a framework for understanding the agency's pragmatics, influenced by its perception, interpretation, communication and adaptation in the world. In the authors' view, pragmatics refers to the agency's ability to face complexity, uncertainty and change and to assume practical objectives that meet its needs and priorities.

The study uses a rich bibliography, specific to the topic addressed.

Based on this and the innovative conception of the authors, a complex and deep analysis of the historical and current issues of ASEAN is presented in the study and the metacybernetic foundation for applying the appropriate analysis tools is consolidated.

The conclusions are relevant and can lead to the improvement of ASEAN activity.

The suggestion we make to the authors is that in the current study they should also introduce a synthesis of the previous theoretical study (part I) to allow the reader a better, autonomous understanding of the entire research

Author Response

We thank you for summarising so well the activities with this paper, and providing a important contribution to improve it .We have integrated your summary into the paper.

Reviewer: The suggestion we make to the authors is that in the current study they should also introduce a synthesis of the previous theoretical study (part I) to allow the reader a better, autonomous understanding of the entire research

Reply: This has been done by introducing material into the introduction relevant to part 1 of the paper. We thank you for this important observation.

Reviewer 3 Report

Comments and Suggestions for Authors

1.     The overall concept of the paper

1.1. Conceptual/logical  structure of the paper

Bearing in mind the rudimentary issues of traditional reviewing, i.e., the aim, hypotheses, etc., it may be concluded that the main aim is defined indirectly:

"Employing the meta-cybernetic modeling framework of Mindset Agency Theory (MAT), a complex adaptive system trait approach, this study explores the underlying attributes of ASEAN to determine its cultural 44 stability. It seeks to provide insights into the possible nature and manifestations of pathologies within the ASEAN framework. It can be read, in cognitive terms – this remark used here, not incidentally, that the Author proposes new models and assumes (hypothesizes) that they may be new useful statistical tools."

The research question at the heart of this study centers on solving pathological problems arising from cultural diversity in complex and dynamic situations. Additionally, the study delves into understanding institutional mechanisms—expressed in terms of agency and its agents—whose characteristics are defined by formative traits. By addressing these attributes, the research aims to offer a comprehensive understanding of the complex problems stemming from cultural diversity and institutional dynamics.

No hypothesis is formulated that is understandable in the case of such research.

The paper is a continuation of the first part of the study. The assessment of this paper is performed with necessary reference to the first part.

The overall formal (logical) concept of the paper is correct. 

1.2. The problem and theoretical background

The research problem presented in the paper is very interesting. However, it has several significant weaknesses. They are apparent when a linguistic/semantic approach is used.

The Authors of both parts are familiar with the broad literature on the complexity of social systems and the strengths and weaknesses of the applications of complexity-related ideas in social studies.

As commonly known, the ideas from broadly defined complexity studies/science/theory embody mathematical models, analogies, and metaphors relating to these models and indigenous qualitative narratives, e.g., social autopoiesis.

The central concept in this paper is a complex adaptive system. The sources of this concept are quoted in part I. In part II, no references are made to the sources. It is not a problem since the readers are directed to part 1.

In both parts, the Authors present intricate considerations based on the polysemous terms with multiple and broad interpretations.

I do not deny some ideas presented in the first part. However, the relations between broadly defined terms presented in Figure 2 in Part I will be challenging to comprehend when a more rigorous semantic approach is applied.

While the first part can be seen as an interesting, although disputable general idea, the application of its results in the second part is so broad that its usefulness may stir some doubts.

I am not a specialist in ASEAN, but the description of this institution is correct. However, declaring it as a complex adaptive system is self-evident, even if the references from part I are taken into account. With such a broad metaphorical interpretation, it can be easily proved that every social system can be described as CAS. It was not the idea of the authors of this concept since all of them – Holland and Gell-Mann, in particular, tried to show a formal, operationalizable sense of adaptation.

I do not delve into the details. It would require preparing a text of a size comparable to the papers under scrutiny.

Another doubt is associated with the relationship between social, cultural, and cognitive attributes of ASEAN treated as a complex adaptive system.

It may sound convincing at a comprehensive metaphorical level (meta-level). However, multiple problems will emerge when the Authors are asked to explain causal links between the metaphors.

I do not insist on simplified operationalizations, but at least a specific order that derives from the rules of formal logic should be maintained. 

2.     Methodology of research

Bearing in mind the paper's conceptual and speculative character, the study's methodology is correct. Its weaknesses derive from the abovementioned epistemological faults.

 3.     The final assessment of the paper

The paper is interesting and constitutes the continuation of research by one of the authors. However, this application merges cognitive, institutional, and functional levels with broadly defined (polysemous) metaphors, which may imply that complex adaptive systems can be applied everywhere.

The paper can be published, but it is necessary to add that it is a specific linguistic exercise in which the Authors apply a more or less provable knowledge about ASEAN with a collection of metaphors and analogies deriving from the vocabulary of complexity studies.

This comment should be made in two situations. Firstly, when referring to part I, the applications of the theoretical concepts in studying ASEAN are shown.

A short comment reflecting the source of the above assessment. In the first part of the study, the term “fractal” is frequently used as a metaphor. For somebody familiar with the mathematical aspects of fractals, the fact that we can impose self-repetitive metaphorical patterns on social and mental models is evident. However, the second aspect of fractals, i.e., fractal dimensionality, is more difficult to apply in a metaphorical sense.

Also, referring to the first part, linking von Foerster with fractals  is interesting, but a question arises as to whether von Foerster himself would go that far. 

The paper can be published only after considering the above suggestions, which shows that the authors are aware of the limitations of using and abusing broadly defined metaphors and analogies.

As another source of inspiration for changing the paper, the authors should consider the possibility of presenting the results of this study to the audience awaiting specific policy recommendations, among whom could be well-trained sociologists, political scientists, and psychologists.

4.     Editorial remarks

The paper is edited carefully. The text is clear. Some minor editorial errors can be found, e.g., lowercase first characters in the last words of the title. References are predominantly relevant and presented meticulously.

Author Response

Thank you for your insightful comments and feedback on our paper. We have carefully considered your suggestions and have made revisions to strengthen the clarity and rigor of our content. In response to your comments regarding the use of metaphors and the development of theoretical frameworks, we have included a new section in the paper that addresses the interplay between metaphor and theory in the realm of complexity theory.

The section highlights the strategic use of metaphors in making abstract concepts more accessible and emphasizes the importance of empirical evidence and methodological rigor in developing substantive theoretical frameworks. We discuss how theories must go beyond metaphorical representation to establish themselves as foundational tools of scientific inquiry, drawing on examples such as contingency theory and stability theory.

Furthermore, we explore the complexities of combining metaphor and theory, explaining how metaphors serve as cognitive devices to comprehend complex abstract ideas and how empirical evidence enriches theoretical frameworks. We discuss the integration of different theoretical perspectives and empirical observations in shaping robust frameworks for inquiries into social and political complexity.

We believe that these additions have addressed your concerns and have enhanced the academic value of the paper. We appreciate your feedback and are confident that the revised paper is now ready for publication.

Thank you for your time and valuable input.